# Joint Relational Database Generation via Graph-Conditional Diffusion Models

**Mohamed Amine Ketata, David Lüdke, Leo Schwinn, Stephan Günnemann**

School of Computation, Information and Technology & Munich Data Science Institute
Technical University of Munich, Germany
Correspondence to: `a.ketata@tum.de`

## Abstract

Building generative models for relational databases (RDBs) is important for many applications, such as privacy-preserving data release and augmenting real datasets. However, most prior works either focus on single-table generation or adapt single-table models to the multi-table setting by relying on autoregressive factorizations and sequential generation. These approaches limit parallelism, restrict flexibility in downstream applications, and compound errors due to commonly made conditional independence assumptions. In this paper, we propose a fundamentally different approach: jointly modeling all tables in an RDB without imposing any table order. By using a natural graph representation of RDBs, we propose the Graph-Conditional Relational Diffusion Model (GRDM), which leverages a graph neural network to jointly denoise row attributes and capture complex inter-table dependencies. Extensive experiments on six real-world RDBs demonstrate that our approach substantially outperforms autoregressive baselines in modeling multi-hop inter-table correlations and achieves state-of-the-art performance on single-table fidelity metrics. Our code is available at `https://github.com/ketatam/rdb-diffusion`.

## 1 Introduction

Relational databases (RDBs), which organize data into multiple interlinked tables, are the most widely used data management system, estimated to store over 70% of the world's structured data [1]. RDBs are used in various domains, including healthcare, finance, education, and e-commerce [2, 3]. However, increasing legal and ethical concerns around data privacy have led to strict regulations that limit access to data containing personal or sensitive information. While these safeguards protect individuals and organizations, they can hinder the development of data-driven technologies that benefit from rich, structured data, thereby slowing scientific progress.

Synthetic data generation has emerged as a promising approach to mitigate this problem by training generative models on private datasets and releasing synthetic samples that preserve key statistical properties without disclosing sensitive information [4, 5]. In addition, synthetic data can enhance fairness, facilitate data augmentation, and support robust downstream analysis [6, 7].

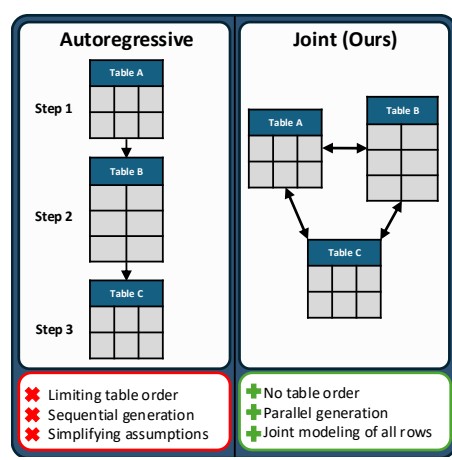

Figure 1: Comparison of autoregressive and joint relational database generation.

39th Conference on Neural Information Processing Systems (NeurIPS 2025).

Despite recent advances in synthetic data generation for single-table settings [8–12], *multi-table* relational database generation remains relatively underexplored. Notably, current approaches share a common limitation: they impose a fixed order on the tables and model their distribution autoregressively [13–16] (*cf.* Figure 1 left). This sequential generation procedure introduces several key limitations. It constrains downstream tasks, such as missing data imputation, prevents parallel sampling, and often struggles with capturing complex long-range dependencies between tables. In addition, prior methods often adopt strong conditional independence assumptions that facilitate learning but sacrifice fidelity or generality [15]. We discuss further related work in Appendix A.

In this paper, we propose a fundamentally different approach: we jointly model all tables in a relational database without relying on a specific table ordering (*cf.* Figure 1 right) and propose the first non-autoregressive generative model for RDBs. To achieve this, we adopt a graph-based representation of RDBs [17], where nodes correspond to individual rows, and edges encode primary-foreign key relationships. By framing RDB generation as a graph generation problem, we leverage random graph generation methods to first sample a graph that preserves node degree distributions of the real graph, then jointly generate all node features using our novel Graph-Conditional Relational Diffusion Model (GRDM). Our main contributions are the following:

- We introduce a new framework that models the joint distribution over all tables in an RDB, removing the need for a predefined table ordering or autoregressive factorization.
- We propose GRDM, the first non-autoregressive generative model for RDBs. It uses a graph-based representation and jointly generates all row attributes.
- We demonstrate that GRDM significantly outperforms autoregressive baselines across six real-world RDBs, especially in capturing long-range dependencies between tables.

## 2 Background

**Relational Databases.** A *Relational Database* (RDB) [18] $(\mathcal{R}, \mathcal{L})$ (*cf.* Figure 2 top) is defined by a set of $m$ *interconnected* tables, called *relations*, $\mathcal{R} = \{R^{(1)}, \ldots, R^{(m)}\}$ and a database schema $\mathcal{L} \subseteq \mathcal{R} \times \mathcal{R}$ that specifies the connections between them. Specifically, a tuple $(R_{child}, R_{parent})$ belongs to $\mathcal{L}$ if rows from $R_{child}$ contain references to rows from $R_{parent}$, forming a *parent-child relationship*. Each table $R^{(i)}$ represents one *entity type* and consists of $n_i$ rows $\{r_1^{(i)}, \ldots, r_{n_i}^{(i)}\}$, where each row $r$ is an instance of this entity and consists of three components [17]:

- **Primary key** $p_r$: a unique identifier for each row within a table.
- **Foreign keys** $\mathcal{K}_r \subseteq \{p_{r'} \mid r' \in R' \text{ and } (R, R') \in \mathcal{L}\}$: a set of primary keys of rows in other tables, that row $r$ points to. We refer to these rows as $r$'s *parents*.
- **Attributes** $\boldsymbol{x}_r$: a set of *columns* (features) describing this specific instance. In this work, we focus on databases with categorical and numerical attributes, which include ordinal, date, time, discrete, and continuous data.

**Synthetic Relational Database Generation.** The goal of synthetic RDB generation is to learn a parameterized distribution $p_\theta \approx p(\mathcal{R})$ to sample a synthetic database $\tilde{\mathcal{R}}$ that adheres to the schema $\mathcal{L}$ and preserves the statistical properties of $\mathcal{R}$. The challenges of this learning problem are twofold:

*(i)* the complex distributions of columns with heterogeneous multi-modal features *(column shapes)*,

*(ii)* the complex correlations between columns from the same table *(intra-table trends)*, as well as columns from different tables *(inter-table trends)* which can be directly *(one-hop)* or indirectly *(multi-hop)* connected by primary-foreign key links.

**Autoregressive Models for RDB Generation.** To learn these distributions and capture their correlations, existing generative models for RDBs rely on a specific table order, often the topological order induced by the schema $\mathcal{L}$, to autoregressively factorize the joint distribution [13, 16, 15, 19, 14]:

$$p(\mathcal{R}) = p(R^{(1)}, \ldots, R^{(m)}) = \prod_{i=1}^{m} p(R^{(i)} \mid \{R^{(i')} \mid i' < i\}).^1$$

---

[1] Some works [16, 19] generate all primary and foreign keys in a first step, then apply a similar autoregressive factorization on the attributes (non-key columns).

While this factorization can, in principle, model the true joint distribution, its inherent sequential nature introduces two key limitations. First, *fixing the table order* limits downstream applications, e.g., missing value imputation, because each table is conditioned only on its predecessors and cannot incorporate information from tables later in the order. Second, the factorization forces the *generation runtime* to scale linearly with the number of tables and prevents parallelization across tables.

Moreover, learning the required conditional probability distributions is practically challenging, particularly for large RDBs with high-dimensional attributes and complex *multi-hop* inter-table dependencies. To make the problem tractable, prior work focused on RDBs with only two tables [16] or resorted to simplifying assumptions such as Markov-like conditional independence assumptions of tables or individual rows given their parents [13–15]. Such assumptions are particularly problematic in this autoregressive setting, where small errors can accumulate across generation steps, causing compounding errors and loss of coherence.

## 3 Joint Modeling of All Tables with Graph-Conditional Diffusion Models

In this work, we introduce the first non-autoregressive generative model for RDBs by directly modeling the *joint* distribution of tables in $\mathcal{R}$. Specifically, by avoiding any table ordering and leveraging the relational structure of the database to jointly model its rows, our approach imposes fewer assumptions and enables models that are more expressive, flexible, and parallelizable. We structure this section as follows: we introduce how to faithfully model RDBs as graphs in Section 3.1, motivate our two-step graph generation procedure in Section 3.2, describe how we generate the graph structure in Section 3.3, present our novel diffusion model, which is the main contribution of this paper, in Section 3.4, and finally, discuss the special case of dimension tables in RDBs in Section 3.5.

### 3.1 Relational Databases as Attributed Heterogeneous Directed Graphs

The relational structure of an RDB is naturally represented as a directed heterogeneous graph that elegantly models its different components. A key advantage of adopting this graph-based view is that we can leverage the vast literature on graph theory and graph machine learning to process relational data. This idea has gained traction in recent representation learning work [20, 17, 21], and has also been explored in generative modeling [16, 19], though existing approaches remain autoregressive. While there are several ways to represent an RDB as a graph [21], we adopt the simple and intuitive formulation from Fey et al. [17], modeling an RDB $(\mathcal{R}, \mathcal{L})$ as a heterogeneous graph in which each row is a node and edges are defined by primary–foreign key links (*cf.* Figure 2 bottom).

Formally, we define the graph as $\mathcal{G} = (\mathcal{V}, \mathcal{E}, \mathcal{X})$, with node set $\mathcal{V}$ representing the rows, edge set $\mathcal{E}$ representing the primary–foreign key connections, and feature set $\mathcal{X}$ representing the attributes. First, we map each row $r \in R^{(i)}$ to a node $v$ of type $i$, resulting in a *heterogeneous* graph. The full node set is given by $\mathcal{V} = \bigcup_{i=1}^{m} \mathcal{V}^{(i)}$, where each $\mathcal{V}^{(i)} = \{v(r) \mid r \in R^{(i)}\}$ contains all nodes of type $i$ and $v(r)$ denotes the node corresponding to row $r$, and $r(v)$ its inverse. Second, we define the edge set as

$$\mathcal{E} = \{(v_1, v_2) \in \mathcal{V} \times \mathcal{V} \mid p_{v_2} \in \mathcal{K}_{v_1}\},$$

where $p_v = p_{r(v)}$ and $\mathcal{K}_v = \mathcal{K}_{r(v)}$ are shorthand for accessing a node's primary and foreign keys.

Each edge represents a primary-foreign key relationship between two rows. We assign each edge a type $(i, j)$, based on the types of its endpoints $v_1$ and $v_2$. Since edges are ordered pairs, the resulting graph is *directed*—a property that is essential for reconstructing the original RDB from its graph representation. Finally, the feature set is defined as $\mathcal{X} = \{\boldsymbol{x}_v \mid v \in \mathcal{V}\}$, where $\boldsymbol{x}_v = \boldsymbol{x}_{r(v)}$ contains the non-key attributes of the corresponding row. Thus,

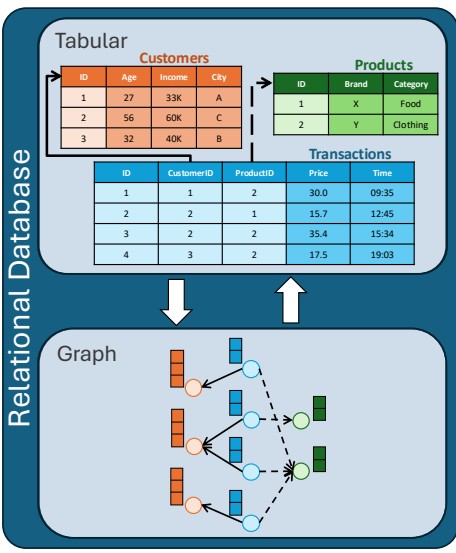

Figure 2: Tabular and graph representations of relational databases. We use different colours and different arrow shapes to depict different node and edge types, respectively.

$\mathcal{X}$ contains all columns in $\mathcal{R}$ that are not primary or foreign keys, while the primary and foreign keys are fully encoded in the (featureless) graph structure $(\mathcal{V}, \mathcal{E})$.

Notably, this graph construction is a *faithful* representation of the original RDB in the sense that one can reconstruct the original RDB $\mathcal{R}$ from the graph $\mathcal{G}$ – up to permutations of the primary keys (thus also of foreign keys). One concrete procedure for this reconstruction is provided in Appendix C.1. In the following, we use $\mathcal{G}$ and $\mathcal{R}$ interchangeably to refer to the graph and RDB representation of the same object. With this, our target data distribution becomes:

$$p(\mathcal{R}) = p(\mathcal{G}) = p(\mathcal{V}, \mathcal{E}, \mathcal{X}). \tag{1}$$

## 3.2 On the Factorization of $p(\mathcal{V}, \mathcal{E}, \mathcal{X})$

To find an efficient and scalable parametrization for $p(\mathcal{V}, \mathcal{E}, \mathcal{X})$, we examine its possible factorizations. First, note that $\mathcal{V}$ is implicitly defined by either $\mathcal{X}$ or $\mathcal{E}$. Hence, we can reduce the problem to three viable factorizations: *(A) one-shot*; $p(\mathcal{V}, \mathcal{E}, \mathcal{X})$, *(B) features-then-edges*; $p(\mathcal{V}, \mathcal{X})p(\mathcal{E}|\mathcal{V}, \mathcal{X})$, or *(C) edges-then-features*; $p(\mathcal{V}, \mathcal{E})p(\mathcal{X}|\mathcal{V}, \mathcal{E})$.

To illustrate the scalability trade-offs among these factorizations, consider an RDB with two tables containing $n$ and $m$ rows, respectively. Note that for factorizations *(A)* and *(B)*, the edges are generated *along with* or *after* the features identifying each node. Thus, they need to consider all $n \times m$ possible edges, which is computationally prohibitive for large $n$ and $m$, and increasingly so for databases with multiple tables. In contrast, factorization *(C)* generates the edges *before* the features, which allows us to sidestep this combinatorial explosion by exploiting the exchangeability of nodes and simply parameterizing the number of nodes and their per edge-type degree for efficient random graph generation similar to Xu et al. [16]. Specifically, factorizing the distribution as

$$p(\mathcal{R}) = p(\mathcal{G}) = p(\mathcal{V}, \mathcal{E})p(\mathcal{X}|\mathcal{V}, \mathcal{E}) \tag{2}$$

corresponds to a two-step process. First, $p(\mathcal{V}, \mathcal{E})$ defines the structure of the attribute-free graph, equivalent to generating the primary and foreign key columns of the RDB. Second, $p(\mathcal{X}|\mathcal{V}, \mathcal{E})$ generates the node features of the graph, or equivalently, the row attributes of the RDB. Note that this can be attained by jointly modeling each node with its neighborhood defined by the graph structure, allowing us to scale to very large databases. In the following, we build up on this factorization: Section 3.3 presents a simple and effective algorithm for modeling $p(\mathcal{V}, \mathcal{E})$, followed by our main contribution, a graph-conditional diffusion framework for modeling $p(\mathcal{X}|\mathcal{V}, \mathcal{E})$, in Section 3.4.

## 3.3 Node Degree-Preserving Random Graph Generation

The featureless graph $(\mathcal{V}, \mathcal{E})$ is a directed, heterogeneous, and $m$-partite graph—edges exist only between nodes of different types, corresponding to rows from different tables. We model $p(\mathcal{V}, \mathcal{E})$ using a simple random graph generation algorithm that preserves the node degree distributions observed in the real graph [22]. We define a node's *indegree* and *outdegree* for a given edge type as the number of incoming and outgoing edges of that type, respectively. Outdegrees are constant per edge type, since each table has a fixed number of foreign key columns. Thus, our goal is to preserve the indegree distribution for each edge type.

The learning phase computes the empirical indegree distributions per edge type from the real graph. Sampling begins with root nodes—those with no outgoing edges. We set the number of root nodes to match the real graph, ensuring a similar overall graph size, but the graph size can also be scaled by a multiplier (see Appendix E.4 for the corresponding experiment). For each root node, we sample its children according to the learned indegree distributions and recursively repeat this process for the sampled child nodes. When a node has multiple parents, this procedure initially produces multiple copies for it. To correct this while preserving the overall indegree distribution, we perform a random matching to merge duplicate nodes across these sets, before sampling their indegree sequence.

## 3.4 Graph-Conditional Relational Diffusion Model (GRDM)

Given the graph structure $(\mathcal{V}, \mathcal{E})$, we are now interested in generating the features $\mathcal{X}$. Since this is now a feature generation task for a fixed graph structure, an interesting analogy to keep in mind is image generation. An image can be viewed as a graph with a simple 2D grid structure, where each pixel represents a node with a 3-dimensional feature vector of color channels. In our case,

we can think about $p(\mathcal{X}|\mathcal{V}, \mathcal{E})$ as generating a very large "image", with a more complex structure and heterogeneous node features. Inspired by the success of diffusion models in image generation and other domains [23–26], and given their unprecedented capacity to model joint distributions, we propose to model $p(\mathcal{X}|\mathcal{V}, \mathcal{E})$ using a diffusion model. Our model can be viewed as a generalization of image diffusion models to graphs with arbitrary structures and node features. In this section, we use $\mathcal{X}^{(0)}$ to denote the original data $\mathcal{X}$, as is common in the diffusion models literature [23].

### 3.4.1 Forward Process

Diffusion models [27, 23] are latent variable models that define a sequence of latent variables $\mathcal{X}^{(1)}, \ldots, \mathcal{X}^{(T)}$ of the same dimensionality as $\mathcal{X}^{(0)}$ through a fixed Markov chain, called the *forward process* or *diffusion process*, that gradually adds noise to data:

$$q(\mathcal{X}^{(1:T)}|\mathcal{X}^{(0)}, \mathcal{V}, \mathcal{E}) = \prod_{t=1}^{T} q(\mathcal{X}^{(t)}|\mathcal{X}^{(t-1)}, \mathcal{V}, \mathcal{E}). \tag{3}$$

While this formulation is very general and allows defining complex forward processes, we choose to add noise *independently* to all nodes at each timestep $t$ (analogous to how pixels are independently noised in image diffusion models), which will enable efficient training of our model:

$$q(\mathcal{X}^{(t)}|\mathcal{X}^{(t-1)}, \mathcal{V}, \mathcal{E}) = \prod_{v \in \mathcal{V}} q(\boldsymbol{x}_v^{(t)}|\boldsymbol{x}_v^{(t-1)}), \tag{4}$$

where $\boldsymbol{x}_v^{(t)}$ is the feature vector of node $v$ at diffusion timestep $t$. In this work, we assume that $\boldsymbol{x}_v$ consists of numerical and categorical features. A common way to model categorical tabular data is multinomial diffusion [28, 8], however, it suffers significant performance overheads and we found it unstable for high-cardinality categorical variables. Therefore, we follow Pang et al. [15] and map categorical variables to continuous space through label encoding (See Appendix C.2 for details) and apply Gaussian diffusion (DDPM [23]) on the unified space: $q(\boldsymbol{x}_v^{(t)}|\boldsymbol{x}_v^{(t-1)}) = \mathcal{N}(\boldsymbol{x}_v^{(t)}; \sqrt{1-\beta_t}\boldsymbol{x}_v^{(t-1)}, \beta_t \boldsymbol{I})$, where $\beta_1, \ldots, \beta_T$ define the variance schedule.

An important consequence of this choice of the forward process, i.e., adding *Gaussian* noise *independently* to all nodes, is that we can sample $\boldsymbol{x}_v^{(t)}$ of an individual node at any timestep $t$ directly from the clean data at time 0, which is key for efficient training of the model. Let $\alpha_t = 1 - \beta_t$ and $\bar{\alpha}_t = \prod_{i=1}^{t} \alpha_i$, then $q(\boldsymbol{x}_v^{(t)}|\mathcal{X}^{(0)}, \mathcal{V}, \mathcal{E}) = q(\boldsymbol{x}_v^{(t)}|\boldsymbol{x}_v^{(0)}) = \mathcal{N}(\boldsymbol{x}_v^{(t)}; \sqrt{\bar{\alpha}_t}\boldsymbol{x}_v^{(0)}, (1-\bar{\alpha}_t)\boldsymbol{I})$.

### 3.4.2 Reverse Process

The generative model is defined as another Markov chain with learned transitions; the *reverse process*,

$$p_\theta(\mathcal{X}^{(0:T)}|\mathcal{V}, \mathcal{E}) = p(\mathcal{X}^{(T)}|\mathcal{V}, \mathcal{E}) \prod_{t=1}^{T} p_\theta(\mathcal{X}^{(t-1)}|\mathcal{X}^{(t)}, \mathcal{V}, \mathcal{E}). \tag{5}$$

This formulation of the reverse process also offers significant modeling flexibility; to predict the less noisy state $\boldsymbol{x}_v^{(t-1)}$ of node $v$, the model can leverage the graph structure $(\mathcal{V}, \mathcal{E})$ and the features of all other nodes at time $t$, $\mathcal{X}^{(t)}$. However, doing so is not scalable for large graphs, but also not necessary because most of the nodes will be irrelevant. Therefore, we propose to condition the denoising of node $v$ at each timestep on the local neighborhood encoding the relations of the database, which achieves our goal of jointly modeling the nodes faithfully to the RDB structure. This is analogous to how image diffusion models based on convolutional neural networks (CNNs) condition the denoising of a pixel on its neighboring pixels [23].

To allow a node to interact not only with its parents, but also with its children, for the diffusion model, we treat the graph $\mathcal{G}$ as undirected by adding reverse edges to all existing edges in $\mathcal{G}$. Formally, let $\mathcal{G}_{v,K}^{(t)} = (\mathcal{V}_{v,K}, \mathcal{E}_{v,K}, \mathcal{X}_{v,K}^{(t)})$ denote the subgraph from $\mathcal{G}$ (now undirected), centered at node $v$, containing all nodes in the $K$-hop neighborhood around $v$, with features at diffusion time $t$. Concretely, for $K = 1$, the nodes in $\mathcal{G}_{v,1}^{(t)}$ correspond to all rows in the database that are referenced by $r(v)$ and those that reference it. Note that we can also bound the worst-case space and time

complexity by sampling a fixed-size set of neighbors as proposed by Hamilton et al. [29]. Formally, we model the following factorization of the reverse process (Equation 5):

$$p(\mathcal{X}^{(T)}|\mathcal{V},\mathcal{E}) = \prod_{v\in\mathcal{V}} p(\boldsymbol{x}_v^{(T)}) \text{ and } p_\theta(\mathcal{X}^{(t-1)}|\mathcal{X}^{(t)},\mathcal{V},\mathcal{E}) = \prod_{v\in\mathcal{V}} p_\theta(\boldsymbol{x}_v^{(t-1)}|\mathcal{G}_{v,K}^{(t)}). \tag{6}$$

A crucial property of our reverse process formulation is the following: due to the multi-step nature of diffusion models, the denoising of node $v$ is not restricted to only conditioning on its $K$-hop neighborhood, but can extend to further away nodes. To see this, let us consider two consecutive timesteps and identify which nodes in $\mathcal{X}^{(t+1)}$ influence $\boldsymbol{x}_v^{(t-1)}$. From Equation 6, we see that from $t \to t-1$, only nodes in $\mathcal{X}_{v,K}^{(t)}$ influence $\boldsymbol{x}_v^{(t-1)}$. However, from $t+1 \to t$, all nodes in $\mathcal{X}_{v,K}^{(t)}$ are in turn influenced by nodes in their respective $K$-hop neighborhood at time $t+1$, which will indirectly influence $\boldsymbol{x}_v^{(t-1)}$. These nodes form the subgraph $\mathcal{G}_{v,2K}^{(t+1)}$, and by induction on the number of diffusion steps, we can show that $\boldsymbol{x}_v^{(t)}$ is influenced by $\mathcal{G}_{v,NK}^{(t+N)}$ at time $t+N$, meaning that our model can in principle model interactions between node pairs that are $\mathcal{O}(TK)$ hops away in the graph during the whole reverse diffusion process. Empirically, this is demonstrated by our model, which uses $K=1$ and effectively captures higher-order correlations (See Section 4 for detailed experiments).

In the case of Gaussian diffusion, the reverse process is also Gaussian: $p(\boldsymbol{x}_v^{(T)}) = \mathcal{N}(\boldsymbol{x}_v^{(T)}; \boldsymbol{0}, \boldsymbol{I})$ and $p_\theta(\boldsymbol{x}_v^{(t-1)}|\mathcal{G}_{v,K}^{(t)}) = \mathcal{N}(\boldsymbol{x}_v^{(t-1)}; \boldsymbol{\mu}_\theta(\mathcal{G}_{v,K}^{(t)}, t), \sigma_t^2 \boldsymbol{I})$, where $\boldsymbol{\mu}_\theta$ is a neural network that learns to approximate the less noisy state of node $v$ given its neighborhood at time $t$, and $\sigma_t$ is a hyperparameter.

### 3.4.3 Training and Sampling

**Training.** Recall that our end goal is to learn a parametrized diffusion model $p_\theta$ for the distribution $p(\mathcal{X}^{(0)}|\mathcal{V},\mathcal{E})$. $p_\theta$ can be learned by minimizing the model's negative log-likelihood (NLL) of the data, $-\log p_\theta(\mathcal{X}^{(0)}|\mathcal{V},\mathcal{E})$. However, the NLL is intractable to compute; we therefore optimize its variational bound [27]:

$$-\log p_\theta(\mathcal{X}^{(0)}|\mathcal{V},\mathcal{E}) \leq \mathbb{E}_q\left[-\log \frac{p_\theta(\mathcal{X}^{(0:T)}|\mathcal{V},\mathcal{E})}{q(\mathcal{X}^{(1:T)}|\mathcal{X}^{(0)},\mathcal{V},\mathcal{E})}\right] =: L(\theta). \tag{7}$$

$L(\theta)$ can be equivalently expressed in terms of conditional forward process posteriors, which are tractable to compute (see Appendix B for a detailed derivation). Furthermore, we follow Ho et al. [23] in parametrizing $\boldsymbol{\mu}_\theta$ as $\boldsymbol{\mu}_\theta(\mathcal{G}_{v,K}^{(t)}, t) = \frac{1}{\sqrt{\alpha_t}}\left(\boldsymbol{x}_v^{(t)} - \frac{\beta_t}{\sqrt{1-\bar{\alpha}_t}}\boldsymbol{\epsilon}_\theta(\mathcal{G}_{v,K}^{(t)}, t)\right)$, where $\boldsymbol{\epsilon}_\theta$ predicts the noise added at time $t$, and using a simplified training objective (details are provided in Appendix B), leading to the following objective that we use to train our model:

$$L_{\text{simple}}(\theta) = \mathbb{E}_{v\sim U(\mathcal{V}), t\sim U(\{1,\dots,T\}), \{\boldsymbol{\epsilon}_{v'}\sim\mathcal{N}(\boldsymbol{0},\boldsymbol{I}) \mid v'\in\mathcal{V}_{v,K}\}}\left[\left\|\boldsymbol{\epsilon}_v - \boldsymbol{\epsilon}_\theta\left(\mathcal{G}_{v,K}^{(t)}, t\right)\right\|^2\right], \tag{8}$$

where $U$ samples an element from its input set uniformly at random. Our model, $\boldsymbol{\epsilon}_\theta$, is trained by minimizing $L_{\text{simple}}(\theta)$ using stochastic gradient descent, where each iteration works as follows: a node $v$ is sampled uniformly at random and is noised along with its $K$-hop neighborhood to the *same* time step $t$, which is also sampled uniformly at random. Given the noisy subgraph $\mathcal{G}_{v,K}^{(t)}$, $\boldsymbol{\epsilon}_\theta$ learns to predict the noise vector $\boldsymbol{\epsilon}_v$ that was added to $v$ (but not the noise vectors added to the other nodes in its neighborhood). Algorithm 1 shows the detailed training procedure.

**Sampling.** To sample from our model, we initialize all node features in the graph with Gaussian noise, $\boldsymbol{x}_v^{(T)} \sim \mathcal{N}(\boldsymbol{0}, \boldsymbol{I}) \; \forall \; v \in \mathcal{V}$. Then, at each time step $t = T, \dots, 1$, all nodes are jointly denoised before moving to the next step as follows:

$$\boldsymbol{x}_v^{(t-1)} = \frac{1}{\sqrt{\alpha_t}}\left(\boldsymbol{x}_v^{(t)} - \frac{\beta_t}{\sqrt{1-\bar{\alpha}_t}}\boldsymbol{\epsilon}_\theta\left(\mathcal{G}_{v,K}^{(t)}, t\right)\right) + \sigma_t \boldsymbol{z}_v \; \forall \; v \in \mathcal{V}, \tag{9}$$

where $\boldsymbol{z}_v \sim \mathcal{N}(\boldsymbol{0}, \boldsymbol{I})$ if $t > 1$ else $\boldsymbol{z}_v = \boldsymbol{0}$. The full sampling procedure is shown in Algorithm 2.

A nice property of our sampling procedure is that at each time step, the denoising step (Equation 9) can be run fully in parallel across all nodes in the graph, allowing us to scale to very large graphs. This is in contrast to prior works [15, 19] that specify an order on the tables and can only start with generating one table, once all its predecessors have already been generated.

### 3.4.4 Model Architecture

The denoising model $\epsilon_\theta$ is a learnable function that maps a graph $\mathcal{G}$ centered at node $v$ and a timestep $t$ to a vector of the same dimension $d$ as $\boldsymbol{x}_v$. Formally, $\epsilon_\theta : \mathbb{G} \times \mathbb{R} \to \mathbb{R}^d$, where $\mathbb{G}$ is the set of graphs. In this work, we use heterogeneous Message-Passing Graph Neural Networks (MP-GNNs) [30, 31] to parameterize $\epsilon_\theta$. Let $(\mathcal{G}, t) \in \mathbb{G} \times \mathbb{R}$ denote the input to $\epsilon_\theta$ with $\mathcal{G} = (\mathcal{V}, \mathcal{E}, \mathcal{X})$. We define initial node embeddings as $\boldsymbol{h}_v^0 = (\boldsymbol{x}_v, t) \ \forall \ v \in \mathcal{V}$. One iteration of heterogeneous message passing consists of updating each node's embedding at iteration $l$, $\boldsymbol{h}_v^l$, based on its neighbors, to get updated embeddings $\boldsymbol{h}_v^{l+1}$, where nodes and edges of different types are treated differently. In Appendix C.4, we present a general formulation of heterogeneous MP-GNNs. In this work, we use the heterogeneous version of the GraphSAGE model [29, 32] with sum-based neighbor aggregation. After $L$ iterations of message passing, we obtain a set of deep node embeddings $\{\boldsymbol{h}_v^L\}_{v \in \mathcal{V}}$. We set $L$ to be the same as the number of hops $K$ used to select $\mathcal{G}_{v,K}$. Finally, to predict the noise vector $\hat{\epsilon}_v$ for the target node $v$, we further pass the final embedding of node $v$ through a node type-specific multi-layer perceptron (MLP) [33]: $\hat{\epsilon}_v = \mathrm{MLP}^{(i)}(\boldsymbol{h}_v^L)$, which is the final output of $\epsilon_\theta$, where $i$ is the type of node $v$.

## 3.5 Dimension Tables

In RDBs, some tables have a unique property: they have relatively few rows compared to other tables, and each row represents a unique real-world entity. An example of such a table is a COUNTRIES table that stores the different countries in the world. This specific type of table is well-known in the RDB literature and is called *dimension* tables [34, 17], which contrasts with so-called *fact* tables. As discussed in [17], dimension tables store contextual information, macro statistics, and/or immutable properties of real-world entities, which makes these tables relatively small. In contrast, fact tables typically store interactions between entities, e.g., transactions or ratings. Since entities interact frequently, fact tables have the majority of rows in an RDB.

In our generative modeling setting, dimension tables present a unique challenge and require special care. Since rows of dimension tables represent unique entities in the real world, there is no clear notion of generalization, and it is unclear how to generate synthetic dimension tables. Therefore, we propose to view dimension tables as metadata, on which we condition the generation of all other tables. This is reminiscent of how possible categories of a categorical random variable are not generated but rather encoded into the generative model. In fact, one can join a dimension table with a table that references it and get a new categorical column (in the joined table) whose possible categories are the entities in the dimension table.

To condition our generative process on dimension tables, we simply avoid adding noise to them and keep them fixed at time $t = 0$ during training and sampling of the diffusion model. In the aforementioned example of the COUNTRIES table, this modified process would only use the information about the countries themselves from the real RDB, while, for example, the information about customers' nationalities is encoded in the edges of the graph and is still stochastically sampled from our model. In our experiments, we define dimension tables as tables that contain at least one categorical column, where each possible category appears only once in that table.

## 4 Experiments

We evaluate our model's performance in synthetic relational database generation, using single-table and multi-table fidelity metrics, including long-range dependency metrics introduced in [15]. We present a comparison of our proposed approach to different state-of-the-art baselines on six real-world databases, followed by a detailed analysis and an ablation study for our model. In Appendix E, we evaluate other aspects, including privacy preservation, missing value imputation, and database size extrapolation.

### 4.1 Experimental Setup

**Real-world Databases.** We use six real-world relational databases in our experiments. Five were used in the evaluation setup in [15]: *Berka* [35], *Instacart 05* [36], *Movie Lens* [37, 38], *CCS* [37], and *California* [39]. In addition, we use the *RelBench-F1* database [40] from RelBench [41], a

recently introduced benchmark for relational deep learning [17]. We chose *F1* because it contains nine tables—more than any of the previous five RDBs—and includes a relatively high number of numerical and categorical columns, compared to other databases in RelBench. The used databases vary in the number of tables, size, maximum depth, number of inter-table connections, and feature complexity. We provide more details on these databases in Appendix D.1. Note that *Berka*, *Instacart 05*, and *RelBench-F1* exhibit the most complex inter-table correlations with up to 3-hop interactions.

**Baselines.** We compare our model to four relational database generation methods from the literature. SDV [13] is a statistical method tailored for RDB synthesis based on Gaussian Copulas. ClavaDDPM [15] is a state-of-the-art diffusion-based model that leverages cluster labels and classifier-guided diffusion models to generate a child table conditioned on its parent. In addition, we adopt two synthesis pipelines used by Pang et al. [15] to provide additional insights: SingleTable and Denorm. SingleTable generates foreign keys based on the real group size distribution, i.e., the distribution of row counts that have the same foreign key, similarly to our random graph generation algorithm. However, it learns and generates each table individually and independently of all other tables. It can be seen as a version of our model with the number of hops $K$ set to 0. Denorm is based on the idea of joining tables first, so it learns and generates the joined table of every connected pair, and then splits it. For both these baselines, we use the same DDPM-based tabular diffusion backbone as our model [23, 8]. For all DDPM-based baselines, we use the same hyperparameters for model architecture and training as our model for a fair comparison. For SDV, we used the default hyperparameters as in [15]. For our model, we set $K = 1$ as we found this setting to achieve a good fidelity-efficiency tradeoff, but better fidelity metrics could be achieved with $K > 1$ at the cost of more compute, which we leave for future investigation. In the context of our experiments, we use "GRDM" to refer to our overall model, i.e., the graph generation component and the diffusion component. More implementation details are provided in Appendix D.2.

**Evaluation Metrics.** To evaluate the quality of the synthetic data in terms of fidelity, we follow [15] and report the following metrics implemented in the SDMetrics package [42]. 1) *Cardinality* compares the distribution of the number of children rows that each parent row has, between the real and synthetic data. We compute this metric for each parent-child pair and report the average across all pairs. 2) *Column Shapes* compares the marginal distributions of individual columns between the real and synthetic data. We compute this metric for every column of each table and report its average. 3) *Intra-Table Trends* measures the correlation between a pair of columns in the same table and computes the similarity in correlation between the real and synthetic data. We report the average across all column pairs of each table. 4) *Inter-Table Trends (k-hop)* is similar to the previous metric but measures the correlation between column pairs from tables at distance $k$, e.g., 1-hop for columns from parent-child pairs. For each possible $k \geq 1$, we report the average across all column pairs that are $k$ hops away from each other. This is the most challenging and most interesting metric because it considers interactions between different tables.

For each of these metrics, we use the Kolmogorov-Smirnov (KS) statistic and the Total Variation (TV) distance to compare distributions of numerical and categorical values, respectively. To compare correlations of column pairs, we use the Pearson correlation coefficient for numerical values and the contingency table for categorical values. All metrics are normalized to lie between 0 (least fidelity) and 100 (highest fidelity). Detailed descriptions of these metric computations are in Appendix D.3.

### 4.2 Evaluation Results

Table 1 reports the fidelity metrics described in Section 4.1 across six databases for all methods, with standard deviations over three random seeds. **GRDM consistently outperforms all baselines on Inter-Table Trends across all databases**, particularly in capturing multi-hop correlations for the most complex RDBs. On *Berka*, which has the most complex schema with up to 3-hop dependencies, GRDM surpasses the best baseline by 14.14% on 3-hop correlations and 11.3% on 2-hop correlations. On *Instacart 05* and *RelBench-F1*, GRDM improves over the best baseline by more than 5.7 times and 12.62% on 2-hop correlations, respectively. For 1-hop correlations, GRDM achieves an average gain of 8.15% compared to the best baselines over all databases. These improvements highlight the effectiveness of our joint modeling approach over the autoregressive methods used by other baselines (with the exception of SingleTable, which treats tables independently). Notably, the fact that our model, which uses neighborhood size $K = 1$, achieves very good multi-hop correlation metrics

Table 1: Comparison of the fidelity metrics described in Section 4.1. The best method in each metric is highlighted, and we report the relative improvement of GRDM compared to the closest baseline. Databases are ordered by complexity. SDV does not support RDBs with more than 5 tables.

| | SDV | SingleTable | Denorm | ClavaDDPM | GRDM (Ours) | Improvement (%) |
|---|---|---|---|---|---|---|
| **Berka** | | | | | | |
| CARDINALITY | | $97.06 _{\pm 0.80}$ | $96.06 _{\pm 1.15}$ | $96.75 _{\pm 0.26}$ | $99.65 _{\pm 0.05}$ | +2.67 |
| COLUMN SHAPES | | $94.58 _{\pm 0.01}$ | $83.28 _{\pm 0.97}$ | $94.60 _{\pm 0.41}$ | $96.84 _{\pm 0.14}$ | +2.37 |
| INTRA-TABLE TRENDS | | $91.72 _{\pm 0.23}$ | $72.12 _{\pm 0.73}$ | $90.53 _{\pm 1.93}$ | $98.23 _{\pm 0.03}$ | +7.10 |
| INTER-TABLE TRENDS (1-HOP) | > 5 tables | $81.77 _{\pm 1.19}$ | $55.77 _{\pm 2.80}$ | $83.79 _{\pm 4.21}$ | $91.41 _{\pm 0.11}$ | +9.09 |
| INTER-TABLE TRENDS (2-HOP) | | $78.09 _{\pm 0.53}$ | $57.68 _{\pm 1.67}$ | $85.87 _{\pm 2.72}$ | $95.57 _{\pm 0.04}$ | +11.30 |
| INTER-TABLE TRENDS (3-HOP) | | $75.56 _{\pm 0.34}$ | $55.59 _{\pm 1.48}$ | $80.98 _{\pm 3.12}$ | $92.43 _{\pm 0.69}$ | +14.14 |
| **Instacart 05** | | | | | | |
| CARDINALITY | | $94.73 _{\pm 0.14}$ | $94.98 _{\pm 0.84}$ | $94.91 _{\pm 1.50}$ | $99.96 _{\pm 0.01}$ | +5.24 |
| COLUMN SHAPES | | $89.30 _{\pm 0.00}$ | $71.83 _{\pm 0.32}$ | $90.18 _{\pm 0.43}$ | $98.39 _{\pm 0.18}$ | +9.10 |
| INTRA-TABLE TRENDS | > 5 tables | $99.70 _{\pm 0.00}$ | $88.74 _{\pm 0.00}$ | $99.68 _{\pm 0.02}$ | $97.59 _{\pm 0.09}$ | -2.12 |
| INTER-TABLE TRENDS (1-HOP) | | $66.93 _{\pm 0.07}$ | $62.58 _{\pm 0.05}$ | $75.84 _{\pm 0.36}$ | $88.49 _{\pm 0.93}$ | +16.68 |
| INTER-TABLE TRENDS (2-HOP) | | $16.22 _{\pm 13.41}$ | $0.00 _{\pm 0.00}$ | $14.40 _{\pm 20.37}$ | $92.95 _{\pm 0.07}$ | +473.06 |
| **RelBench-F1** | | | | | | |
| CARDINALITY | | $94.94 _{\pm 1.06}$ | $93.39 _{\pm 1.56}$ | $94.04 _{\pm 2.33}$ | $98.44 _{\pm 0.16}$ | +3.69 |
| COLUMN SHAPES | | $95.93 _{\pm 0.16}$ | $89.25 _{\pm 0.28}$ | $95.19 _{\pm 0.75}$ | $95.71 _{\pm 0.05}$ | -0.23 |
| INTRA-TABLE TRENDS | > 5 tables | $95.80 _{\pm 0.29}$ | $90.30 _{\pm 0.56}$ | $94.71 _{\pm 0.63}$ | $95.00 _{\pm 0.11}$ | -0.84 |
| INTER-TABLE TRENDS (1-HOP) | | $79.61 _{\pm 0.65}$ | $65.60 _{\pm 0.45}$ | $88.19 _{\pm 0.27}$ | $90.85 _{\pm 0.15}$ | +3.02 |
| INTER-TABLE TRENDS (2-HOP) | | $74.10 _{\pm 2.71}$ | $62.21 _{\pm 0.38}$ | $83.17 _{\pm 1.39}$ | $93.67 _{\pm 0.03}$ | +12.62 |
| **Movie Lens** | | | | | | |
| CARDINALITY | | $98.99 _{\pm 0.16}$ | $98.87 _{\pm 0.26}$ | $98.79 _{\pm 0.13}$ | $98.80 _{\pm 0.36}$ | -0.19 |
| COLUMN SHAPES | | $99.19 _{\pm 0.00}$ | $78.03 _{\pm 0.17}$ | $99.11 _{\pm 0.09}$ | $97.22 _{\pm 0.48}$ | -1.99 |
| INTRA-TABLE TRENDS | > 5 tables | $98.56 _{\pm 0.01}$ | $57.33 _{\pm 0.10}$ | $98.52 _{\pm 0.05}$ | $95.25 _{\pm 0.35}$ | -3.36 |
| INTER-TABLE TRENDS (1-HOP) | | $92.72 _{\pm 0.09}$ | $77.45 _{\pm 1.93}$ | $92.11 _{\pm 2.12}$ | $94.34 _{\pm 0.89}$ | +1.75 |
| **CCS** | | | | | | |
| CARDINALITY | $74.36 _{\pm 8.40}$ | $99.37 _{\pm 0.16}$ | $26.70 _{\pm 0.20}$ | $98.96 _{\pm 0.79}$ | $99.79 _{\pm 0.03}$ | +0.42 |
| COLUMN SHAPES | $69.04 _{\pm 4.38}$ | $95.20 _{\pm 0.00}$ | $79.29 _{\pm 0.13}$ | $92.64 _{\pm 3.93}$ | $97.30 _{\pm 0.36}$ | +2.21 |
| INTRA-TABLE TRENDS | $94.84 _{\pm 1.00}$ | $98.96 _{\pm 0.00}$ | $86.60 _{\pm 0.14}$ | $97.75 _{\pm 1.70}$ | $94.82 _{\pm 1.20}$ | -4.18 |
| INTER-TABLE TRENDS (1-HOP) | $21.74 _{\pm 9.62}$ | $51.62 _{\pm 0.22}$ | $57.77 _{\pm 0.69}$ | $72.65 _{\pm 8.10}$ | $85.38 _{\pm 3.03}$ | +17.52 |
| **California** | | | | | | |
| CARDINALITY | $71.45 _{\pm 0.00}$ | $99.89 _{\pm 0.04}$ | $99.87 _{\pm 0.02}$ | $98.99 _{\pm 0.69}$ | $99.96 _{\pm 0.01}$ | +0.07 |
| COLUMN SHAPES | $72.32 _{\pm 0.00}$ | $99.51 _{\pm 0.04}$ | $94.99 _{\pm 0.02}$ | $98.76 _{\pm 0.27}$ | $98.75 _{\pm 0.00}$ | -0.76 |
| INTRA-TABLE TRENDS | $50.23 _{\pm 0.00}$ | $98.69 _{\pm 0.08}$ | $94.17 _{\pm 0.01}$ | $97.65 _{\pm 0.39}$ | $97.35 _{\pm 0.02}$ | -1.36 |
| INTER-TABLE TRENDS (1-HOP) | $54.89 _{\pm 0.00}$ | $92.96 _{\pm 0.05}$ | $87.24 _{\pm 0.10}$ | $95.34 _{\pm 0.48}$ | $96.16 _{\pm 0.05}$ | +0.86 |

confirms that long-range dependencies can be effectively captured through the diffusion process, as a node's receptive field increases with each diffusion step (see Section 3.4.2 for a detailed discussion on this point). On single-table metrics, GRDM performs comparably to the best methods and even outperforms them on some databases.

## 4.3 Ablation Studies

To gain further insights into our model and understand the performance of its different components, we conduct two ablation studies that analyze the capacity of our random graph generation algorithm (Section 3.3) and the importance of the joint modeling aspect. For brevity, we only report results on the three databases with the most complex schema, *Berka*, *Instacart 05*, and *RelBench-F1*, shown in Table 2.

**Real Graph Structure.** To better understand the modeling capacity of our graph generation algorithm described in Section 3.3, we perform an ablation study by replacing the graph sampled using this algorithm with the

Table 2: Comparison of our default setting from Table 1 with using the real graph structure instead of sampling it and with a sequential version of our model.

| | GRDM | Real Graph | Sequential |
|---|---|---|---|
| **Berka** | | | |
| CARDINALITY | $99.65 _{\pm 0.05}$ | $100.0 _{\pm 0.00}$ | $99.65 _{\pm 0.05}$ |
| COLUMN SHAPES | $96.84 _{\pm 0.14}$ | $96.91 _{\pm 0.06}$ | $38.97 _{\pm 3.65}$ |
| INTRA-TABLE TRENDS | $98.23 _{\pm 0.03}$ | $98.19 _{\pm 0.06}$ | $52.10 _{\pm 2.98}$ |
| INTER-TABLE TRENDS (1-HOP) | $91.41 _{\pm 0.11}$ | $91.50 _{\pm 0.17}$ | $28.53 _{\pm 3.84}$ |
| INTER-TABLE TRENDS (2-HOP) | $95.57 _{\pm 0.04}$ | $95.90 _{\pm 0.05}$ | $52.01 _{\pm 2.55}$ |
| INTER-TABLE TRENDS (3-HOP) | $92.43 _{\pm 0.69}$ | $94.45 _{\pm 0.56}$ | $49.45 _{\pm 4.14}$ |
| **Instacart 05** | | | |
| CARDINALITY | $99.96 _{\pm 0.01}$ | $100.0 _{\pm 0.00}$ | $99.96 _{\pm 0.01}$ |
| COLUMN SHAPES | $98.39 _{\pm 0.18}$ | $98.84 _{\pm 0.03}$ | $67.51 _{\pm 3.93}$ |
| INTRA-TABLE TRENDS | $97.59 _{\pm 0.09}$ | $98.57 _{\pm 0.04}$ | $89.33 _{\pm 2.27}$ |
| INTER-TABLE TRENDS (1-HOP) | $88.49 _{\pm 0.93}$ | $91.80 _{\pm 0.11}$ | $63.32 _{\pm 2.01}$ |
| INTER-TABLE TRENDS (2-HOP) | $92.95 _{\pm 0.07}$ | $92.85 _{\pm 0.10}$ | $13.41 _{\pm 3.64}$ |
| **RelBench-F1** | | | |
| CARDINALITY | $98.44 _{\pm 0.16}$ | $100.0 _{\pm 0.00}$ | $98.44 _{\pm 0.16}$ |
| COLUMN SHAPES | $95.71 _{\pm 0.05}$ | $96.78 _{\pm 0.05}$ | $94.83 _{\pm 0.09}$ |
| INTRA-TABLE TRENDS | $95.00 _{\pm 0.11}$ | $95.84 _{\pm 0.06}$ | $95.01 _{\pm 0.75}$ |
| INTER-TABLE TRENDS (1-HOP) | $90.85 _{\pm 0.15}$ | $93.12 _{\pm 0.30}$ | $90.82 _{\pm 0.37}$ |
| INTER-TABLE TRENDS (2-HOP) | $93.67 _{\pm 0.03}$ | $96.10 _{\pm 0.23}$ | $87.73 _{\pm 0.86}$ |

ground-truth graph from the RDB and using the same diffusion model from the previous experiments to generate the attributes. The results are shown in the second column of Table 2. Using the real graph achieves a perfect relational structure fidelity (*cf.* Cardinality metric) and provides an upper-bound on the performance achievable by any graph generative model. The results show that our model achieves close performance to the real graph setting, confirming that our graph generation procedure can effectively capture the relational structure of the RDBs, with the small gap indicating that there is room for improvement by improving our graph generation procedure, which we leave for future research. This gap is smaller or even nonexistent on the simpler databases, as shown in Table 6 in the Appendix.

**Effect of Joint Modeling.** To isolate the impact of our joint modeling approach, we implemented an autoregressive variant of our model that retains all other components (e.g., graph representation, GNN architecture). Instead of modeling $p(\mathcal{X}|\mathcal{V}, \mathcal{E})$ directly, it factorizes the distribution as $\prod_{i=1}^{m} p(\mathcal{X}^{(i)} \mid \{\mathcal{X}^{(i')} \mid i' < i\})$, where $\mathcal{X}^{(i)}$ denotes features of node type $i$. As shown in the third column of Table 2, this version performs significantly worse than our proposed approach, not only on inter-table correlations, but also on single-table metrics. This aligns with observations from [15], which introduced latent cluster variables to mitigate the difficulty of learning conditional distributions with noisy and high-dimensional conditing space. Note that the sequential generation is significantly slower than with the joint model. Concretely, on Berka, the autoregressive model was more than 2.5 times slower than its joint counterpart.

## 5 Conclusion

We introduced GRDM, the first non-autoregressive generative model for RDBs. GRDM represents RDBs as graphs by mapping rows to nodes and primary–foreign key links to edges. It first generates a graph structure using a degree-preserving random graph model, then jointly generates all node features–i.e., row attributes–via a novel diffusion model that conditions each node's denoising on its $K$-hop neighborhood. This enables modeling long-range dependencies due to the iterative nature of diffusion models. GRDM consistently outperformed baselines on six real-world RDBs, particularly in capturing long-range correlations.

**Limitations and Future Work.** This work lays the foundation for scalable and expressive generative models for RDBs, and opens several avenues for future research. First, while our graph generation method is simple and effective, more tailored approaches could better capture the structure of relational data. Second, although we use a straightforward graph representation, alternatives–such as modeling certain tables (e.g., transactions or reviews) as attributed edges–may be more appropriate for specific tasks. Third, while this work focuses on unconditional generation, GRDM can be extended to support downstream tasks and serve as a foundational model for relational data. Finally, equipping GRDM with differential privacy guarantees could further increase its practical adoption, especially in sensitive domains.

## Broader Impact

This paper introduces a new approach for generating synthetic relational databases. Some positive impacts of our work include enabling more accurate and powerful synthetic data generation, which promises to tackle issues with privacy-preserving data sharing and advancing scientific research. However, such models can be used to generate fake content that can be misused. However, we strongly believe that the benefits significantly outweigh the small chance of misuse.

## Acknowledgments

This project was funded by SAP SE. It is also supported by the DAAD programme Konrad Zuse Schools of Excellence in Artificial Intelligence, sponsored by the Federal Ministry of Education and Research. We thank Pedro Henrique Martins, Maximilian Schambach, Sirine Ayadi, and Tim Beyer for their valuable feedback.

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

# A  Related Work

While the focus of this work is on relational database generation, our method draws on ideas from various fields, which we briefly review in this section. As RDBs are composed of multiple tables, tabular data generation methods are a major building block of RDB generative models. We start by reviewing both lines of work. Then, we review some recent efforts to apply graph machine learning to RDBs, and finally discuss existing works for general graph generation.

**Tabular data generation.**  For single-table generation, early work [43] applied Bayesian networks to model the distribution of tabular data with graphical models. CTGAN [44] is a seminal work that applied Generative Adversarial Networks (GANs) [45] to tabular data generation. Recently, with the rise of diffusion models, more diffusion-based models have been proposed to model tabular data. Notably, TabDDPM [8] was the first to apply denoising diffusion probabilistic models to the tabular data domain. TabSyn [10] adopts the powerful idea of latent diffusion using a transformer-based variational autoencoder. More recently, TabDiff [11] proposes learning feature-specific node schedules and applying masked diffusion to model categorical variables.

**Relational Database Generation.**  The Synthetic Data Vault (SDV) [13] laid the foundation of this field. It uses the Gaussian copula process to model every parent-child relationship. PrivLava [14] applies the framework of differential privacy to generate RDBs. It uses graphical models with latent variables to capture the inter-table correlations. ClavaDDPM [15] is a recent diffusion-based model that introduced cluster-based latent variables and uses classifier-guided diffusion models to generate the rows of a child table conditioned on its parent. Xu et al. [16] and Hudovernik [19] adopt a graph view of the RDB and model the autoregressive factorization of the tables' distribution. Note that all prior methods for RDB generation also follow a similar autoregressive factorization.

**Relational Deep Learning.**  Recently, the idea of representing RDBs as graphs and applying graph machine learning methods to process them has gained traction by the introduction of Relational Deep Learning (RDL) [17] and the release of a benchmark of real-world databases (RelBench) [41]. While we use a similar graph representation, the sole focus of these works is on predictive tasks on RDBs, e.g., to predict user churn or future purchases. In contrast, we are interested in generative modeling of RDBs, which is a more general and challenging problem.

**Graph generation.**  By framing RDB generation as a graph generation problem, our work draws on a large literature on graph generation. Early work in this direction builds on the classic problem of finding graphs with a given degree sequence [22]. Closest to our setup is the setting of large graph generation, typically applied to social networks or citation networks. GAE and VGAE [46] extend autoencoders and variational autoencoders the the graph setting. NetGAN [47] applies GANs for graph structure generation by sequentially generating random walks. Li et al. [48] introduces a diffusion model for large graph generation. However, all these works focus on general graphs like social networks, which typically have a complex graph structure but rather simple features. In contrast, in our case, the graphs induced by relational databases exhibit a rather simple graph structure with specific properties (like m-partite) but very complex attributes with heterogeneous and multi-modal features. This distinguishes our work from all these prior works and presents new challenges.

# B    Detailed Loss Derivation

We provide a detailed derivation for the NLL loss following [27, 23]. For notation brevity, we drop the conditioning on $\mathcal{V}, \mathcal{E}$ in the following:

$$
\begin{aligned}
&- \log p_\theta(\mathcal{X}^{(0)}) \\
&\leq \mathbb{E}_q\left[-\log \frac{p_\theta(\mathcal{X}^{(0:T)})}{q(\mathcal{X}^{(1:T)}|\mathcal{X}^{(0)})}\right] \\
&= \mathbb{E}_q\left[-\log p(\mathcal{X}^{(T)}) - \sum_{t=1}^T \log \frac{p_\theta(\mathcal{X}^{(t-1)}|\mathcal{X}^{(t)})}{q(\mathcal{X}^{(t)}|\mathcal{X}^{(t-1)})}\right] \\
&= \mathbb{E}_q\left[-\sum_{v\in\mathcal{V}} \log p(\boldsymbol{x}_v^{(T)}) - \sum_{t=1}^T \sum_{v\in\mathcal{V}} \log \frac{p_\theta(\boldsymbol{x}_v^{(t-1)}|\mathcal{G}_{v,K}^{(t)})}{q(\boldsymbol{x}_v^{(t)}|\boldsymbol{x}_v^{(t-1)})}\right] \\
&= \mathbb{E}_q\left[-\sum_{v\in\mathcal{V}} \log p(\boldsymbol{x}_v^{(T)}) - \sum_{t=2}^T \sum_{v\in\mathcal{V}} \log \frac{p_\theta(\boldsymbol{x}_v^{(t-1)}|\mathcal{G}_{v,K}^{(t)})}{q(\boldsymbol{x}_v^{(t)}|\boldsymbol{x}_v^{(t-1)})} - \sum_{v\in\mathcal{V}} \log \frac{p_\theta(\boldsymbol{x}_v^{(0)}|\mathcal{G}_{v,K}^{(1)})}{q(\boldsymbol{x}_v^{(1)}|\boldsymbol{x}_v^{(0)})}\right] \\
&= \mathbb{E}_q\left[-\sum_{v\in\mathcal{V}} \log p(\boldsymbol{x}_v^{(T)}) - \sum_{t=2}^T \sum_{v\in\mathcal{V}} \log \frac{p_\theta(\boldsymbol{x}_v^{(t-1)}|\mathcal{G}_{v,K}^{(t)})}{q(\boldsymbol{x}_v^{(t-1)}|\boldsymbol{x}_v^{(t)}, \boldsymbol{x}_v^{(0)})} \cdot \frac{q(\boldsymbol{x}_v^{(t-1)}|\boldsymbol{x}_v^{(0)})}{q(\boldsymbol{x}_v^{(t)}|\boldsymbol{x}_v^{(0)})}\right. \\
&\qquad\qquad \left. - \sum_{v\in\mathcal{V}} \log \frac{p_\theta(\boldsymbol{x}_v^{(0)}|\mathcal{G}_{v,K}^{(1)})}{q(\boldsymbol{x}_v^{(1)}|\boldsymbol{x}_v^{(0)})}\right] \\
&= \mathbb{E}_q\left[-\sum_{v\in\mathcal{V}} \log \frac{p(\boldsymbol{x}_v^{(T)})}{q(\boldsymbol{x}_v^{(T)}|\boldsymbol{x}_v^{(0)})} - \sum_{t=2}^T \sum_{v\in\mathcal{V}} \log \frac{p_\theta(\boldsymbol{x}_v^{(t-1)}|\mathcal{G}_{v,K}^{(t)})}{q(\boldsymbol{x}_v^{(t-1)}|\boldsymbol{x}_v^{(t)}, \boldsymbol{x}_v^{(0)})} - \sum_{v\in\mathcal{V}} \log p_\theta(\boldsymbol{x}_v^{(0)}|\mathcal{G}_{v,K}^{(1)})\right] \\
&= \mathbb{E}_q\left[\sum_{v\in\mathcal{V}} \underbrace{D_{\mathrm{KL}}\left(q(\boldsymbol{x}_v^{(T)}|\boldsymbol{x}_v^{(0)})\|p(\boldsymbol{x}_v^{(T)})\right)}_{L_v^{(T)}} + \sum_{t=2}^T \sum_{v\in\mathcal{V}} \underbrace{D_{\mathrm{KL}}\left(q(\boldsymbol{x}_v^{(t-1)}|\boldsymbol{x}_v^{(t)}, \boldsymbol{x}_v^{(0)})\|p_\theta(\boldsymbol{x}_v^{(t-1)}|\mathcal{G}_{v,K}^{(t)})\right)}_{L_v^{(t-1)}}\right. \\
&\qquad\qquad \left. - \sum_{v\in\mathcal{V}} \underbrace{\log p_\theta(\boldsymbol{x}_v^{(0)}|\mathcal{G}_{v,K}^{(1)})}_{L_v^{(0)}}\right] \\
&= \mathrm{const.} + \mathbb{E}_q\left[\sum_{t=2}^T \sum_{v\in\mathcal{V}} \underbrace{D_{\mathrm{KL}}\left(q(\boldsymbol{x}_v^{(t-1)}|\boldsymbol{x}_v^{(t)}, \boldsymbol{x}_v^{(0)})\|p_\theta(\boldsymbol{x}_v^{(t-1)}|\mathcal{G}_{v,K}^{(t)})\right)}_{L_v^{(t-1)}} - \sum_{v\in\mathcal{V}} \underbrace{\log p_\theta(\boldsymbol{x}_v^{(0)}|\mathcal{G}_{v,K}^{(1)})}_{L_v^{(0)}}\right] \\
&=: L(\theta)
\end{aligned}
$$

The advantage of this reformulation is that the forward process posteriors needed to compute the loss are tractable when conditioned on $\boldsymbol{x}_v^{(0)}$:

$$
q(\boldsymbol{x}_v^{(t-1)}|\boldsymbol{x}_v^{(t)}, \boldsymbol{x}_v^{(0)}) = \mathcal{N}(\boldsymbol{x}_v^{(t-1)}; \tilde{\boldsymbol{\mu}}_t(\boldsymbol{x}_v^{(t)}, \boldsymbol{x}_v^{(0)}), \tilde{\beta}_t \boldsymbol{I}), \tag{10}
$$

with

$$
\tilde{\boldsymbol{\mu}}_t(\boldsymbol{x}_v^{(t)}, \boldsymbol{x}_v^{(0)}) = \frac{1}{\sqrt{\alpha_t}}\left(\boldsymbol{x}_v^{(t)}(\boldsymbol{x}_v^{(0)}, \boldsymbol{\epsilon}) - \frac{\beta_t}{\sqrt{1-\bar{\alpha}_t}}\boldsymbol{\epsilon}\right), \boldsymbol{\epsilon} \sim \mathcal{N}(\boldsymbol{0}, \boldsymbol{I}) \tag{11}
$$

and

$$
\tilde{\beta}_t = \frac{1 - \bar{\alpha}_{t-1}}{1 - \bar{\alpha}_t}\beta_t \tag{12}
$$

The term $L_v^{(t-1)}$, which is a KL divergence between two Gaussians, can now be written as:

$$L_v^{(t-1)} = \mathbb{E}_q \left[ \frac{1}{2\sigma_t^2} \left\| \tilde{\boldsymbol{\mu}}_t(\boldsymbol{x}_v^{(t)}, \boldsymbol{x}_v^{(0)}) - \boldsymbol{\mu}_\theta(\mathcal{G}_{v,K}^{(t)}, t) \right\|^2 \right] + \text{const.}$$

$$= \mathbb{E}_q \left[ \frac{1}{2\sigma_t^2} \left\| \frac{1}{\sqrt{\alpha_t}} \left( \boldsymbol{x}_v^{(t)}(\boldsymbol{x}_v^{(0)}, \boldsymbol{\epsilon}_v) - \frac{\beta_t}{\sqrt{1 - \bar{\alpha}_t}} \boldsymbol{\epsilon}_v \right) - \boldsymbol{\mu}_\theta(\mathcal{G}_{v,K}^{(t)}, t) \right\|^2 \right] + \text{const.} \tag{13}$$

This suggests a new parametrization of $\boldsymbol{\mu}_\theta$, which predicts the noise instead:

$$\boldsymbol{\mu}_\theta(\mathcal{G}_{v,K}^{(t)}, t) = \frac{1}{\sqrt{\alpha_t}} \left( \boldsymbol{x}_v^{(t)} - \frac{\beta_t}{\sqrt{1 - \bar{\alpha}_t}} \boldsymbol{\epsilon}_\theta(\mathcal{G}_{v,K}^{(t)}, t) \right) \tag{14}$$

With this, $L_v^{(t-1)}$ is as follows:

$$L_v^{(t-1)} = \mathbb{E}_q \left[ \frac{\beta_t^2}{2\sigma_t^2 \alpha_t(1 - \bar{\alpha}_t)} \left\| \boldsymbol{\epsilon}_v - \boldsymbol{\epsilon}_\theta(\mathcal{G}_{v,K}^{(t)}, t) \right\|^2 \right] \tag{15}$$

Ho et al. [23] found that training works even better with a simplified objective that ignores the weighting term in $L_v^{(t-1)}$ and treats $L_v^{(0)}$ similarly to all $L_v^{(t-1)}$

$$L_{\text{simple}}(\theta) = \mathbb{E}_{t \sim U(\{1,\ldots,T\}), v \sim U(\mathcal{V}), q} \left[ \left\| \boldsymbol{\epsilon}_v - \boldsymbol{\epsilon}_\theta \left( \mathcal{G}_{v,K}^{(t)}, t \right) \right\|^2 \right] \tag{16}$$

Sampling from $q$ requires sampling a set of noise vectors, one for each node in $v$'s neighborhood: $\{\boldsymbol{\epsilon}_{v'} \sim \mathcal{N}(\boldsymbol{0}, \boldsymbol{I}) \mid v' \in \mathcal{V}_{v,K}\}$, and computing the noisy features in $\mathcal{G}_{v,K}^{(t)}$. The final loss can be written as:

$$L_{\text{simple}}(\theta) = \mathbb{E}_{t \sim U(\{1,\ldots,T\}), v \sim U(\mathcal{V}), \{\boldsymbol{\epsilon}_{v'} \sim \mathcal{N}(\boldsymbol{0}, \boldsymbol{I}) \mid v' \in \mathcal{V}_{v,K}\}} \left[ \left\| \boldsymbol{\epsilon}_v - \boldsymbol{\epsilon}_\theta \left( \mathcal{G}_{v,K}^{(t)}, t \right) \right\|^2 \right] \tag{17}$$

## C  Additional Method Details

### C.1  Database Reconstruction from its Graph Representation

In Section 3.1, we presented the graph representation of the RDB that we use for generation. An important property of this representation is that it is invertible, i.e., one can reconstruct the RDB from its graph representation after generating the graph. One way to perform this reconstruction is as follows:

1. For each node type $i$, assign a unique primary key $p_v \in \{1, \ldots, n_i\}$ to each node $v$ of type $i$. Nodes of the same type should have different primary keys.

2. For each edge $(v_1, v_2) \in \mathcal{E}$, add primary key $p_{v_2}$ to the set of foreign keys $\mathcal{K}_{v_1}$.

3. For each node type $i$, construct table $R^{(i)}$ by stacking rows of the form $(p_v, \mathcal{K}_v, \boldsymbol{x}_v)$ for every node $v \in \mathcal{V}^{(i)}$.

### C.2  Gaussian Diffusion for Categorical Variables

In Section 3.4.1, we discussed that our diffusion model applies Gaussian diffusion both to categorical and numerical features by first mapping categorical variables to continuous space through label encoding. Specifically, given a categorical feature with $n$ distinct values $\{c_1, \ldots, c_n\}$, we apply a label encoding scheme $E : \{c_1, \ldots, c_n\} \to \{0, \ldots, n-1\}$ that maps each unique category $c_i$ to a unique integer value. Note that this process is invertible, allowing us to map generated values back to the categorical space.

## C.3 Training and Sampling Algorithms

---
**Algorithm 1** Training
---
1: **repeat**
2:      Sample $v \sim \text{Uniform}(\mathcal{V})$
3:      $(\mathcal{V}_{v,K}, \mathcal{E}_{v,K}, \mathcal{X}_{v,K}^{(0)}) \leftarrow \text{SELECT}_K(v; \mathcal{V}, \mathcal{E}, \mathcal{X}^{(0)})$     $\triangleright$ $K$-hop neighborhood around node $v$
4:      Sample $t \sim \text{Uniform}(\{1, \ldots, T\})$
5:      Sample $\{\boldsymbol{\epsilon}_{v'} \sim \mathcal{N}(\mathbf{0}, \boldsymbol{I}) \mid v' \in \mathcal{V}_{v,K}\}$
6:      $\mathcal{X}_{v,K}^{(t)} \leftarrow \{\sqrt{\bar{\alpha}_t}\boldsymbol{x}_{v'}^{(0)} + \sqrt{1 - \bar{\alpha}_t}\boldsymbol{\epsilon}_{v'} \mid v' \in \mathcal{V}_{v,K}\}$
7:      $\hat{\boldsymbol{\epsilon}}_v \leftarrow \boldsymbol{\epsilon}_\theta(\underbrace{(\mathcal{V}_{v,K}, \mathcal{E}_{v,K}, \mathcal{X}_{v,K}^{(t)})}_{\mathcal{G}_{v,K}^{(t)}}, t)$
8:      Take gradient descent step on $\nabla_\theta \|\boldsymbol{\epsilon}_v - \hat{\boldsymbol{\epsilon}}_v\|^2$
9: **until** converged

---

---
**Algorithm 2** Sampling
---
1: $\mathcal{X}^{(T)} \leftarrow \{\boldsymbol{x}_v^{(T)} \sim \mathcal{N}(\mathbf{0}, \boldsymbol{I}) \mid v \in \mathcal{V}\}$
2: **for** $t = T, \ldots, 1$ **do**
3:      $\mathcal{Z} \leftarrow \{\boldsymbol{z}_v \sim \mathcal{N}(\mathbf{0}, \boldsymbol{I})$ if $t > 1$, else $\boldsymbol{z}_v = \mathbf{0} \mid v \in \mathcal{V}\}$
4:      $\mathcal{X}^{(t-1)} \leftarrow \left\{\frac{1}{\sqrt{\alpha_t}}\left(\boldsymbol{x}_v^{(t)} - \frac{\beta_t}{\sqrt{1-\bar{\alpha}_t}}\boldsymbol{\epsilon}_\theta\left(\mathcal{G}_{v,K}^{(t)}, t\right)\right) + \sigma_t \boldsymbol{z}_v \mid v \in \mathcal{V}\right\}$
5: **end for**
6: **return** $\mathcal{X}^{(0)}$

---

## C.4 Heterogeneous Message-Passing Graph Neural Networks

A common way to process graph-structured data with deep learning models is using Message-Passing Graph Neural Networks (MP-GNNs). Since we are dealing with heterogeneous graphs, we leverage a heterogeneous message passing formulation, which supports a wide range of GNN architectures. Let $(\mathcal{G}, t) \in \mathbb{G} \times \mathbb{R}$ denote the input to $\boldsymbol{\epsilon}_\theta$ with $\mathcal{G} = (\mathcal{V}, \mathcal{E}, \mathcal{X})$. We define initial node embeddings as $\boldsymbol{h}_v^0 = (\boldsymbol{x}_v, t) \ \forall \ v \in \mathcal{V}$. One iteration of message passing consists of computing updated node embeddings $\{\boldsymbol{h}_v^{l+1}\}_{v \in \mathcal{V}}$ from the embeddings at the previous iteration $\{\boldsymbol{h}_v^l\}_{v \in \mathcal{V}}$.

Given a node $v$, let $\phi(v)$ denote its type in the graph and let $E_v$ denote the set of distinct edge types adjacent to $v$. One iteration (or layer) of heterogeneous message passing updates $v$'s embedding $\boldsymbol{h}_v^l$ as follows. First, a distinct message is computed for each edge type:

$$\forall e \in E_v, \ \boldsymbol{m}_{v,e}^{l+1} = \{\!\!\{ g_e(\boldsymbol{h}_w^l) \mid w \in \mathcal{N}_e(v) \}\!\!\},$$

where $\mathcal{N}_e(v)$ is the $e$-specific neighborhood of node $v$. Then, these messages are combined into a single unified message:

$$\boldsymbol{m}_v^{l+1} = \{\!\!\{ f_e(\boldsymbol{m}_{v,e}^{l+1}) \mid e \in E_v \}\!\!\},$$

which is used to update the node embedding:

$$\boldsymbol{h}_v^{l+1} = f_{\phi(v)}(\boldsymbol{h}_v^l, \boldsymbol{m}_v^{l+1}).$$

$g_e$, $f_e$ and $f_{\phi(v)}$ are arbitrary differentiable functions with learnable parameters and $\{\!\!\{\cdot\}\!\!\}$ is a permutation-invariant set aggregator such as mean, max, etc.. Different GNN architectures can be obtained through specific choices of these functions.

# D   Experimental Details

## D.1   Datasets

| | # Tables | # Foreign Key Pairs | Depth | Total # Attributes | # Rows in Largest Table | # Dimension Tables |
|---|---|---|---|---|---|---|
| *Berka* | 8 | 8 | 4 | 41 | $1,056,320$ | 1 |
| *Intacart 05* | 6 | 6 | 3 | 12 | $1,616,315$ | 3 |
| *RelBench-F1* | 9 | 13 | 3 | 21 | $28,115$ | 3 |
| *Movie Lens* | 7 | 6 | 2 | 14 | $996,159$ | 0 |
| *CCS* | 5 | 4 | 2 | 11 | $383,282$ | 1 |
| *California* | 2 | 1 | 2 | 15 | $1,690,642$ | 0 |

Table 3: Database Details

## D.2   Implementation Details

### D.2.1   GRDM hyperparameters and training

**Architecture.** For the GNN, we use the heterogeneous version of the GraphSAGE model [29] with the number of layers set to the number of hops $K$ from the diffusion model. In all our experiments, we set $K = 1$. We use sum-based aggregation and a hidden dimension of 256 for the GNN.

In addition to the GNN, GRDM contains an MLP for each node type. These MLPs have the same architecture as those used in [15] and follow the TabDDPM implementation [8]. We use layers of sizes $512, 1024, 1024, 1024, 1024, 512$, if the corresponding table has more than $10,000$ rows, and use a smaller MLP with layers of sizes $512, 1024, 1024, 512$ otherwise. This, in total, leads to a smaller number of MLP parameters than the models used in [15], which we use as our baselines. All input and output dimensions of the GNN and MLP are adapted to the data dimensions. We also incorporate timestep information by encoding timesteps into sinusoidal embeddings, which are then added to the model's input.

**Diffusion hyperparameters.** We follow [15] and set the diffusion timesteps $T = 2000$ and use cosine scheduler for the noise schedule.

**Training hyperparemters.** We also follow [15] and use the AdamW optimizer with learning rate $6e\text{-}4$ and weight decay $1e\text{-}5$. We use $100,000$ training steps for California and $200,000$ on all other databases. We use a batch size of $4096$ on all databases.

### D.2.2   Baselines

For all diffusion baselines, we use the same set of hyperparameters, including diffusion timesteps, MLP architecture, and training from [15]. In particular, the only difference to our model is that the baselines use larger MLPs, which gives them an advantage over our model in terms of the total number of parameters.

For SDV, we use the default setting of their HMASynthesizer, which uses a Gaussian Copula synthesizer.

### D.2.3   Hardware

All experiments are run using a single NVIDIA A100-PCIE-40GB GPU.

## D.3   Detailed Metrics Computation

First, we define the distribution similarity measures that are used across the different metrics in this work.

For numerical variables, we use the complement of the Kolmogorov-Smirnov (KS) statistic. The KS statistic between two numerical distributions is defined as the maximum difference between their respective cumulative density functions (CDFs).

$$\text{KS} = \sup_x \left| F_{real}(x) - F_{syn}(x) \right|,$$

where $F_{real}$ and $F_{syn}$ denote the CDFs of the real and synthetic variables, respectively. We always report the scaled complement $(1 - \text{KS}) * 100$ such that a higher score means better quality.

For categorical variables, we use the complement of the Total Variation (TV) distance. The TV distance between two categorical distributions compares their probabilities as follows:

$$\text{TV} = \frac{1}{2} \sum_{\omega \in \Omega} \left| R_\omega - S_\omega \right|,$$

where $\Omega$ is the set of all categories, $R_\omega$ and $S_\omega$ are the real and synthetic probabilities for a category $\omega \in \Omega$. We always report the scaled complement $(1 - \text{TV}) * 100$ such that a higher score means better quality.

We now present how each metric is computed in detail.

**Cardinality.** For each parent-child pair in the database, we compute the cardinality of each parent row, i.e., the number of children that each parent row has. This defines a numerical distribution for the real and synthetic data, for which we compute the complement of the KS statistic. The cardinality score is then the average across all parent-child pairs in the database.

**Column Shapes.** For each column in every table of the database, we measure the similarity between the column's marginal distribution in the real and synthetic databases. For numerical values, we use the complement of the KS statistic, while for categorical values, we use the complement of the TV distance. The Column Shapes metric is the average across all columns in the database.

**Intra-Table Trends.** For each pair of columns within the same table, we compute different correlation metrics depending on whether the columns are numerical or categorical (for a pair of numerical and categorical columns, we first discretize the numerical column into bins, then treat both columns as categorical). The Intra-Table Trends is the average of the correlation scores across all column pairs from the same table in the database.

For a pair of numerical columns, $A$ and $B$, we compute the Pearson correlation coefficient defined as

$$\rho_{A,B} = \frac{Cov(A, B)}{\sigma_A \sigma_B},$$

where $Cov$ is the covariance, $\sigma_A$ and $\sigma_B$ are the standard deviations of $A$ and $B$, respectively. We compute this metric for both the real and synthetic pair, yielding $R_{A,B}$ and $S_{A,B}$, respectively. We report the normalized score $\left( 1 - \frac{\left| S_{A,B} - R_{A,B} \right|}{2} \right) * 100$

For a pair of categorical columns, $A$ and $B$, we compute the normalized contingency table, which consists of the proportion of rows that have each combination of categories in $A$ and $B$, i.e., how often each pair of categories co-occur in the same row. We compute this matrix on both the real and synthetic pairs, yielding $R_{A,B}$ and $S_{A,B}$, respectively. We then compute the difference between them using the TV distance as follows:

$$score = \frac{1}{2} \sum_{\alpha \in \Omega_A} \sum_{\beta \in \Omega_B} \left| S_{A,B}(\alpha, \beta) - R_{A,B}(\alpha, \beta) \right|,$$

where $\Omega_A$ and $\Omega_B$ are the set of possible categories of columns $A$ and $B$, respectively. We again report the normalized score $(1 - score) * 100$.

**Inter-Table Trends ($k$-hop).** This metric is similar to the Column Pair Trends, but instead of computing the correlation on column pairs from the same table, it computes it on column pairs from tables that are within $k$ hops from each other. For example, 1-hop considers all parent-child pairs, 2-hop considers a column in some table and its correlation with columns from its parent's parent or child's child, etc. For every possible hop $k$, we report the average across all possible pairs. In practice, this is implemented by denormalizing the pair of tables into a single one and then computing the Column Pair Trends metric.

# E  Additional Experiments

## E.1  Privacy Sanity Check

In order to evaluate the privacy preservation of our model and to test if any memorization is happening, we follow prior work [8, 15] and compute the distance to closest record (DCR) [49] between the generated and the training database. Specifically, for each synthetic sample, we get the minimum L1 distance to the real records. We report the mean DCR, i.e., the average of these distances over all generated samples. Since different tables can have different feature scales and ranges, we also report the DCR values from the holdout set to the training set, which serve as "ground-truth generalizations".

We use four tables from different databases that contain sensitive information (but have been anonymized for these public benchmarks), such as census data containing household and individual information, sensitive financial information from a bank, and order information from an e-commerce platform. The results, provided in Table 4, show that our model yields higher DCR values than the baseline ClavaDDPM (i.e., our model is more private) across all tables and it even yields higher DCR values than the holdout set on 3 out of 4 tables, while being close on the 4th one. These results suggest that our model is able to generate new records from the underlying distribution and is not simply memorizing the training database. Note that DCR values need to be considered along with fidelity metrics (see Table 1), since, e.g., random noise can yield high DCR values.

Table 4: Mean DCR values on selected tables.

|  | California (Household) | California (Individual) | Berka (Transaction) | Instacart 05 (Order) |
|---|---|---|---|---|
| MEAN DCR - HOLDOUT SET | 0.094 | 0.206 | 0.012 | **0.012** |
| MEAN DCR - CLAVADDPM | 0.104 | 0.177 | 0.019 | 0.007 |
| MEAN DCR - GRDM (OURS) | **0.119** | **0.259** | **0.028** | 0.009 |

## E.2  Missing Value Imputation

To assess the performance of GRDM in the task of missing value imputation, we design a new experiment on the California database, which consists of two tables: a parent table and a child table. First, we split the database into a training set and a holdout set based on the parent table, and use the training set to train the diffusion model in the same way as presented in the main text. Next, we perform experiments on the holdout set by conditioning the generation on specific sections and using the same pre-trained model to inpaint the missing sections without any modification to the pre-trained model nor to the sampling procedure [50]. We report the metrics comparing the generated database with the holdout set in Table 5. We use three different strategies to select the missing parts of the database:

1. Masking entire tables (parent or child table).
2. Masking entire rows from both tables with different rates.
3. Masking single cells (individual attributes of rows) from both tables with different rates.

The results show that GRDM consistently maintains good performance across the different masking settings, which again highlights the effectiveness of our proposed joint modeling approach in capturing the complex distributions of RDBs. Note that the first column of Table 5 (Unconditional) is obtained by evaluating against the real test set, which explains the minor differences to the results reported in Table 1.

Table 5: Results of missing value imputation experiments on the California dataset.

|  | Unconditional | Parent Table Missing | Child Table Missing | 50% Rows Missing | 75% Rows Missing | 50% Cells Missing | 75% Cells Missing |
|---|---|---|---|---|---|---|---|
| CARDINALITY | 99.78 | 100.0 | 100.0 | 100.0 | 100.0 | 100.0 | 100.0 |
| COLUMN SHAPES | 99.03 | 99.26 | 99.37 | 99.22 | 99.05 | 98.70 | 98.10 |
| INTRA-TABLE TRENDS | 97.75 | 98.65 | 98.56 | 98.36 | 97.91 | 96.34 | 94.44 |
| 1-HOP | 97.34 | 96.48 | 97.16 | 97.66 | 97.17 | 97.20 | 95.96 |

Table 6: Comparison of our default setting from Table 1 with using the real graph structure instead of sampling it on the databases missing from Table 2.

| | GRDM | Real Graph |
|---|---|---|
| **Movie Lens** | | |
| CARDINALITY | $98.80_{\pm 0.36}$ | $100.0_{\pm 0.00}$ |
| COLUMN SHAPES | $97.22_{\pm 0.48}$ | $97.31_{\pm 0.33}$ |
| INTRA-TABLE TRENDS | $95.25_{\pm 0.35}$ | $95.50_{\pm 0.35}$ |
| 1-HOP | $94.34_{\pm 0.89}$ | $94.71_{\pm 0.82}$ |
| **CCS** | | |
| CARDINALITY | $99.79_{\pm 0.03}$ | $100.0_{\pm 0.00}$ |
| COLUMN SHAPES | $97.30_{\pm 0.36}$ | $97.08_{\pm 0.31}$ |
| INTRA-TABLE TRENDS | $94.82_{\pm 1.20}$ | $93.35_{\pm 2.02}$ |
| 1-HOP | $85.38_{\pm 3.03}$ | $87.87_{\pm 2.70}$ |
| **California** | | |
| CARDINALITY | $99.96_{\pm 0.01}$ | $100.0_{\pm 0.00}$ |
| COLUMN SHAPES | $98.75_{\pm 0.00}$ | $98.75_{\pm 0.01}$ |
| INTRA-TABLE TRENDS | $97.35_{\pm 0.02}$ | $97.36_{\pm 0.02}$ |
| 1-HOP | $96.16_{\pm 0.05}$ | $96.14_{\pm 0.08}$ |

## E.3 Detailed Evaluation of the Random Graph Generation Procedure

In this section, we provide a more fine-grained evaluation of our graph generation algorithm described in Section 3.3. Specifically, we compare the graphs generated using this algorithm with the ground-truth graphs from the original RDBs beyond the Cardinality metric reported in the main text. The cardinality metric compares the node indegree distributions between the real and synthetic databases. In this experiment, we also compute metrics for higher-order structural properties, such as the correlation between node indegrees from different edge types (Correlations), e.g., the correlation between the number of actors in a movie and the number of ratings it received. In addition, we report inter-table indegree correlations (1-Hop, 2-Hop), e.g., the correlation between the number of stores in a specific region and the number of purchases these stores get.

The results, presented in Table 7, show that, even though our graph generation algorithm only models first-order indegree distributions,

Table 7: Higher-order structural properties. Cardinality compares first-order node indegrees, Correlations compares the correlation between node indegrees from different edge types, and 1-Hop, 2-Hop compare inter-table indegree correlations.

| | ClavaDDPM | GRDM (Ours) |
|---|---|---|
| **Berka** | | |
| CARDINALITY | $96.75_{\pm 0.26}$ | $99.65_{\pm 0.05}$ |
| CORRELATIONS | $93.07_{\pm 2.67}$ | $93.64_{\pm 0.18}$ |
| 1-HOP | $98.35_{\pm 0.08}$ | $98.57_{\pm 0.13}$ |
| 2-HOP | $97.87_{\pm 0.46}$ | $99.17_{\pm 0.51}$ |
| **Instacart 05** | | |
| CARDINALITY | $94.91_{\pm 1.50}$ | $99.96_{\pm 0.01}$ |
| CORRELATIONS | $76.51_{\pm 0.34}$ | $95.06_{\pm 0.09}$ |
| 1-HOP | $97.99_{\pm 0.16}$ | $100.0_{\pm 0.00}$ |
| **Movie Lens** | | |
| CARDINALITY | $98.79_{\pm 0.13}$ | $98.80_{\pm 0.36}$ |
| CORRELATIONS | $97.02_{\pm 0.37}$ | $94.56_{\pm 0.31}$ |
| **CCS** | | |
| CARDINALITY | $98.96_{\pm 0.79}$ | $99.79_{\pm 0.03}$ |
| CORRELATIONS | $98.00_{\pm 0.25}$ | $95.73_{\pm 0.18}$ |

it achieves good higher-order performance. This suggests that the used databases do not exhibit strong correlations between different types of relationships and that these relationships are, to a large extent, independent, which our approach can capture very well (which was the main motivation behind our choice in the early stages of this project). However, more complex approaches may be required for more complex databases, which is an interesting research direction, albeit orthogonal to the main focus of this work, namely the joint modeling of tables in RDBs.

## E.4 Database Size Extrapolation

Our graph generation algorithm, presented in Section 3.3, enables the generation of graphs of arbitrary sizes. We leverage this property to evaluate the performance of GRDM when increasing the database size. We use different multipliers of the original database size and generate databases with up to more than 20 million rows (10x multiplier) using the same diffusion model. For this experiment, we use smaller models than the ones used in the main text (specifically, smaller MLPs) for efficiency purposes. The results, shown in Tables 8 and 9 for two databases (California and MovieLens, respectively),

highlight that the performance does not drop when increasing the database size. This can be explained by the use of the GNN in the denoising model, which operates on local subgraphs that are roughly invariant to the overall size of the RDB. Note that all RDBs were generated within less than 2 hours on a single GPU.

Table 8: Performance of GRDM on the California database when increasing its size by different multipliers.

| Size multiplier (total number of rows) | 0.5x (1.037M) | 1x (2.076M) | 2x (4.154M) | 5x (10.381M) | 10x (20.755M) |
|---|---|---|---|---|---|
| **California** | | | | | |
| CARDINALITY | 99.89 | 99.96 | 99.96 | 99.99 | 99.97 |
| COLUMN SHAPES | 97.67 | 97.67 | 97.69 | 97.68 | 97.67 |
| INTRA-TABLE TRENDS | 95.12 | 95.12 | 95.15 | 95.14 | 95.14 |
| 1-HOP | 94.67 | 94.64 | 94.64 | 94.63 | 94.64 |

Table 9: Performance of GRDM on the MovieLens database when increasing its size by different multipliers.

| Size multiplier (total number of rows) | 0.5x (0.588M) | 1x (1.221M) | 2x (2.501M) | 5x (6.270M) | 10x (12.414M) |
|---|---|---|---|---|---|
| **Movie Lens** | | | | | |
| CARDINALITY | 98.69 | 98.87 | 99.10 | 99.63 | 99.65 |
| COLUMN SHAPES | 97.67 | 97.56 | 97.86 | 97.72 | 97.83 |
| INTRA-TABLE TRENDS | 95.38 | 94.99 | 95.58 | 95.34 | 95.42 |
| 1-HOP | 95.17 | 95.35 | 95.57 | 95.62 | 95.63 |

