# OpenReview forum: "Joint Relational Database Generation via Graph-Conditional Diffusion Models"
_NeurIPS.cc/2025/Conference — NeurIPS 2025 poster_

### Official Review · Reviewer_FJA8 · 2025-06-19

**Clarity:** 4
**Significance:** 3
**Originality:** 3
**Rating:** 4
**Confidence:** 4

**Summary:**

The authors propose a generative model for relational tables represented as graphs, where rows are treated as vertices and connected via key–foreign key relationships. The proposed Graph-Conditional Relational Diffusion Model (GRDM) is built on graph neural networks and demonstrates good performance.

**Questions:**

Q1: Can the authors provide concrete evidence or real-world applications to demonstrate the significance of the problem being addressed?

Q2: Can the authors clarify the use cases where augmenting real datasets is necessary? In practice, relational tables often contain large volumes of records.

Q3: Regarding multi-modal features, can the authors specify which data types are considered in this work? In relational databases, data such as images and audio are typically stored as BLOBs and managed externally.

Q4: Can the authors explain why key–foreign key relationships were the focus, and whether other types of relationships (e.g., many-to-many) were considered?

**Ethical Concerns:**

["NO or VERY MINOR ethics concerns only"]

**Final Justification:**

I maintain a score of Borderline accept.

**Limitations:**

No.

**Quality:**

3

**Strengths And Weaknesses:**

Strengths:

S1: Proposed techniques are methodologically sound.

S2: Experimental results demonstrate that GRDM achieves good performance.

S3: The paper reads well.


Weaknesses:

W1: The motivation for the problem is not sufficiently articulated.

W2: The paper only considers key–foreign key relationships; other relationship types, such as many-to-many, are not discussed.

W3: The limitations of the proposed approach are not directly discussed.

---

> ### Author Rebuttal · Authors · 2025-07-30
>
> We thank the reviewer for their valuable feedback. In the following, we address the reviewer's concerns in detail. We will include all the new discussions in the camera-ready version.
>
> ---
> ## [W1, Q1] Motivation and real-world applications
> In the following, we provide concrete real-world applications of the problem of synthetic relational database generation. Note that good generative models for RDBs are a prerequisite for all the following tasks.
>
> * **Privacy-preserving research and development**. Real relational databases typically contain sensitive information (e.g., personal data, financial records). Sharing them with third parties (e.g., LLM providers) or using them for product development can raise serious legal issues, e.g., on the basis of the European Union's General Data Protection Regulation (GDPR). Synthetic relational databases can be used to develop new features or valuable insights, without exposure of real user data. As a concrete example, the SyntheticMass project created a synthetic population of Massachusetts residents (based on EHR schemas) to support research and development without patient risk. For an evaluation of our model's privacy preservation capacity, we refer to our response to Reviewer s1Pb ([W1, W2, Q2] Privacy preservation and generalization).
> * **Data augmentation for machine learning models**. Many real-world machine learning tasks leverage relational databases (e.g., fraud detection, recommendation systems). Synthetic RDBs can help augment underrepresented groups to improve the fairness of the downstream models or simulate rare events to improve their robustness.
> * **Training foundation models on structured data**. There have been recent efforts to build foundation models for structured data [1,2]. However, training such models requires huge amounts of relational databases, which can be accelerated using generative models for RDBs.
> * **Adapting generative models for predictive tasks**. A pre-trained diffusion model for RDBs can be easily adapted to perform conditional generation without the need for re-training [3]. This allows us to leverage a single model for different downstream prediction tasks, such as regression or classification, without the need for training different models for each task, thereby saving cost and time. For concrete results on how our model can be adapted to such cases, we refer to our response to Reviewer gCht ([Q1] Missing and incomplete tables at test time)
>
> We will update the motivation section in the camera-ready version to include these real-world applications.
>
> ---
> ## [W2, Q4] Many-to-many relationships
> We confirm that many-to-many relationships were considered in our work. In fact, **4 out of the 6 used databases contain this type of relationship (Instacart 05, RelBench-F1, MovieLens, and CCS)**. As pointed out by the reviewer, relational databases cannot directly implement many-to-many relationships due to the fixed number of foreign keys a table can have. To overcome this, a so-called join table is typically introduced, which acts as an intermediary, storing the relationships between the two connected tables, essentially breaking down the many-to-many relationship into two one-to-many relationships.
> Examples of this type of relationship from our databases include ratings (from MovieLens), which connect multiple users with multiple movies, and transactions (from CCS), which connect multiple customers with multiple products.
> Since we represent databases as graphs, this implementation offers a unique advantage: the resulting graphs are simple (no multi-edges) and all the attributes are stored as node features instead of edge features (e.g., rating scores), allowing us to use standard GNN architectures out-of-the-box.
>
> We refer to Table 1 in the paper for results on these databases. We will clarify this point in the camera-ready version.
>
> ---
> ## [W3] Limitations
> We kindly point to the last paragraph in the paper (right after the conclusion) for a detailed discussion on the limitations of our approach and promising future research directions.
>
> ---
> ## [Q2] Augmenting real datasets
> Real datasets can be biased towards a specific population or only model common events. In such cases, using generative models to augment real datasets can improve the fairness of the dataset and improve downstream performance on rare edge cases (if, e.g., used to train downstream machine learning models).
> As a concrete example, in the context of healthcare research, synthetic data can be generated to simulate rare diseases. Or in the case of recommendation systems, synthetic data can be used to improve the performance of downstream models on underrepresented groups.
>
> ---
> ## [Q3] Multi-modal features
> In this work, we focus on categorical and numerical data types, which cover different data formats, such as ordinal data, enums, boolean, datetime, and any discrete or continuous data.
> For high-dimensional data such as images, audio, or text, a natural idea would be to model low-dimensional latent representations for these data types as numerical features and directly apply our framework to generate them. Then, a suitable decoder can be applied to map back to the respective original data space.
>
> ---
> ## References
>
> [1] Wang, Yanbo, et al. Griffin: Towards a Graph-Centric Relational Database Foundation Model.
>
> [2] Fey, Matthias, et al. KumoRFM: A Foundation Model for In-Context Learning on Relational Data.
>
> [3] Lugmayr, Andreas, et al. Repaint: Inpainting using denoising diffusion probabilistic models.
>
> ---
>
> We hope our response can address the reviewer's concerns, and we are happy to answer any further questions during the discussion period.

---

> > ### Comment · Reviewer_FJA8 · 2025-08-05
> >
> > Thank authors for the explanation. I maintain my score.

---

> > > ### Author Response · Authors · 2025-08-07
> > >
> > > Dear Reviewer FJA8,
> > >
> > > Thanks a lot for acknowledging our rebuttal. As the discussion period is coming to an end in less than 2 days, we are happy to answer any outstanding or new questions.
> > >
> > > Best regards,
> > >
> > > Authors

---

### Official Review · Reviewer_eXTm · 2025-07-02

**Clarity:** 3
**Significance:** 3
**Originality:** 3
**Rating:** 4
**Confidence:** 3

**Summary:**

This paper addresses the problem of generating synthetic relational databases (RDBs). It identifies key limitations in prior autoregressive approaches, such as sequential generation and fixed table ordering, which can fail to capture complex inter-table dependencies. To overcome this, the authors propose the Graph-Conditional Relational Diffusion Model (GR-DDM), a novel two-stage framework. In the first stage, the model generates the high-level relational graph structure, defining the number of rows for each table and the primary-foreign key relationships between them. In the second stage, conditioned on this generated graph, a diffusion model is used to jointly and non-autoregressively generate the feature values for all rows across all tables. The authors claim this approach better models the complex joint distribution of RDBs and achieves state-of-the-art performance.

**Questions:**

The paper is very promising. My evaluation would be further improved if the authors could clarify the following points:

- On the Richness of Graph Generation: Your node degree-preserving algorithm for graph generation is a clear and reasonable approach. However, have you considered if this is sufficient for capturing more complex, higher-order structural correlations in real-world databases (e.g., correlations between the counts of different relationships)? A brief discussion on the limitations of this structural generation approach would be valuable.

- Scalability: How does GR-DDM scale with the complexity of the database, i.e., the number of tables, columns, and total rows?

- Handling Hard Constraints: How does your framework enforce hard database constraints, such as unique primary keys, or ensure that foreign keys in one table correctly map to existing primary keys in another?

**Ethical Concerns:**

["NO or VERY MINOR ethics concerns only"]

**Final Justification:**

The response addressed my concerns of model scalability, therefore I will maintain the positive rating.

**Limitations:**

yes

**Paper Formatting Concerns:**

no format concerns

**Quality:**

3

**Strengths And Weaknesses:**

Strengths:

- Novel and Elegant Framework (Originality & Significance): The core contribution—a two-stage process that decouples graph structure generation from row content generation—is highly original and compelling. This non-autoregressive approach is a significant departure from prior work and has the potential to better capture holistic inter-table dependencies. The problem of high-fidelity RDB generation is very important, and this work presents a major potential advancement.

- Use of Powerful Generative Models (Quality): The adoption of diffusion models for the content generation stage is a modern and powerful choice. Conditioning these state-of-the-art models on a relational graph structure is a technically sound and interesting research direction.

- Clarity of Presentation: The paper's core idea is communicated very effectively. The abstract is clear, and figures provide an excellent and intuitive understanding of the proposed method and its advantages. The pipeline, including the statistical graph generation method, is well-explained.

Weaknesses:

- Potential Simplifications in Graph Structure Generation (Quality): The paper now clearly explains the graph generation algorithm, which preserves node indegree distributions from the real graph. While this is a reasonable and data-driven approach, it may be a simplification of more complex real-world graph structures. For instance, this method may not capture higher-order structural properties, such as correlations between different types of relationships for a given entity. A discussion on the limitations of only preserving first-order degree distributions would strengthen the paper.

- Scalability Concerns (Quality): While the parallel generation of rows is an advantage, the overall scalability of the method is not discussed. Diffusion models can be computationally intensive, and conditioning on an entire, potentially massive, relational graph could pose significant memory and computational challenges. The paper would be stronger if it addressed how the framework scales to databases with a large number of tables or millions of rows.

- Handling Constraints (Quality): Real-world databases are rich in discrete data and are governed by hard constraints (e.g., unique primary keys). The paper does not specify how the proposed diffusion model framework handles these fundamental characteristics of tabular data, which is a crucial aspect for practical usability.

---

> ### Author Rebuttal · Authors · 2025-07-30
>
> We thank the reviewer for their valuable feedback. In the following, we address the reviewer's concerns in detail. We will include all the new experiments and discussions in the camera-ready version.
>
> ---
> ## [W1, Q1] Richness of Graph Generation
> The cardinality metric reported in the paper compares the node indegree distributions between the real and synthetic databases. As per the reviewer's request, we now also compare higher-order structural properties, such as the correlation between node indegrees from different edge types (**Correlations**), e.g., the correlation between the number of actors in a movie and the number of ratings it received. In addition, we report inter-table indegree correlations (**1-Hop, 2-Hop**), e.g., the correlation between the number of stores in a specific region and the number of purchases these stores get.
>
> The results show that, even though our graph generation algorithm only models first-order indegree distributions, **it achieves good higher-order performance**. This suggests that the used databases do not exhibit strong correlations between different types of relationships and that these relationships are, to a large extent, independent, which our approach can capture very well (which was the main motivation behind our choice in the early stages of this project). However, more complex approaches may be required for more complex databases, which is an interesting research direction, albeit orthogonal to the main focus of this work, namely the joint modeling of tables in RDBs.
>
> **Higher-order structural properties. Cardinality compares first-order node indegree distributions, Correlations compares the correlation between node indegrees from different edge types, and 1-Hop, 2-Hop compare inter-table indegree correlations:**
> ||ClavaDDPM|GRDM (Ours)|
> |-|-|-|
> |**Berka**|
> |Cardinality|96.75 ± 0.26|99.7 ± 0.07|
> |Correlations|93.07 ± 2.67|93.64 ± 0.18|
> |1-Hop|98.35 ± 0.08|98.57 ± 0.13|
> |2-Hop|97.87 ± 0.46|99.17 ± 0.51|
> |**Instacart 05**|
> |Cardinality|94.91 ± 1.50|99.96 ± 0.01|
> |Correlations|76.51 ± 0.34|95.06 ± 0.09|
> |1-Hop|97.99 ± 0.16|100.0 ± 0.00|
> |**Movie Lens**|
> |Cardinality|98.79 ± 0.13|98.8 ± 0.36|
> |Correlations|97.02 ± 0.37|94.56 ± 0.31|
> |**CCS**|
> |Cardinality|98.96 ± 0.79|99.79 ± 0.03|
> |Correlations|98.0 ± 0.25|95.73 ± 0.18|
>
> To further evaluate whether our graph generation approach is sufficient to capture the complexity of the RDBs, we perform a new ablation by replacing the sampled graph with the ground-truth graph and using the same diffusion model to generate the attributes. The ground-truth graph achieves a perfect relational structure fidelity (cf. Cardinality metric) and provides an upper-bound on the performance achievable by graph generative models.
> The results show that our model achieves **very similar performance to the ground-truth graph setting**, showing that **our graph generation procedure can effectively capture the relational structure of the RDBs**.
>
> ||Sampled Graph (Ours)|Ground-truth Graph|
> |-|-|-|
> |**Berka**|||
> |Cardinality|99.70 ± 0.07|100.0 ± 0.00|
> |Column Shapes|96.90 ± 0.07|97.00 ± 0.14|
> |Intra-Table Trends|98.21 ± 0.05|98.24 ± 0.06|
> |1-Hop|92.94 ± 0.06|93.16 ± 0.02|
> |2-Hop|96.04 ± 0.20|96.67 ± 0.14|
> |3-Hop|93.13 ± 0.47|94.47 ± 0.94|
> |**Movie Lens**|||
> |Cardinality|98.80 ± 0.36|100.0 ± 0.00|
> |Column Shapes|98.22 ± 0.05|98.25 ± 0.10|
> |Intra-Table Trends|96.24 ± 0.24|96.31 ± 0.15|
> |1-Hop|95.60 ± 0.17|94.19 ± 2.12|
> |**California**|||
> |Cardinality|99.96 ± 0.01|100.0 ± 0.00|
> |Column Shapes|99.15 ± 0.02|99.15 ± 0.01|
> |Intra-Table Trends|98.00 ± 0.01 |98.02 ± 0.02|
> |1-Hop|97.68 ± 0.01 |97.68 ± 0.02|
>
> ---
> ## [W2, Q2] Scalability to larger databases
> The databases used in our evaluation cover a wide range of sizes. The number of tables ranges from 2 to 9, the number of columns from 11 to 41, and the total number of rows goes up to 2 million (see Table 3 in the paper for details).
> To evaluate the scalability of our model beyond the setting presented in the paper, we leverage the capability of our graph generation algorithm to generate graphs of arbitrary sizes. We use different multipliers of the original database size and generate databases with up to more than **20 million rows** (10x multiplier) using the same diffusion model.
> The table below shows that **our model maintains its good performance across different scales** and that the runtime scales roughly linearly with the database size, **generating the largest database (> 20 million rows) in less than 2 hours on a single GPU**. Both these results can be explained by the use of our GNN in the denoising model, which operates on local subgraphs that are roughly invariant to the overall size of the RDB.
> Note that we used the same batch size for all scales, but the generation can be even faster with larger batch sizes, as the denoising of all rows is performed in parallel at each denoising step.
>
> |Size multiplier (number of rows)|0.5x (1.037M)|1x (2.076M)|2x (4.154M)|5x (10.381M)|10x (20.755M)|
> |-|-|-|-|-|-|
> |**California**|
> |Cardinality|99.89|99.96|99.96|99.99|99.97|
> |Column Shapes|97.67|97.67|97.69|97.68|97.67|
> |Intra-Table Trends|95.12|95.12|95.15|95.14|95.14|
> |1-Hop|94.67|94.64|94.64|94.63|94.64
> |**Runtime (min)**|**7**|**12**|**21**|**58**|**105**|
>
> |Size multiplier (number of rows)|0.5x (0.588M)|1x (1.221M)|2x (2.501M)|5x (6.270M)|10x (12.414M)|
> |-|-|-|-|-|-|
> |**Movie Lens**|
> |Cardinality|98.69|98.87|99.10|99.63|99.65|
> |Column Shapes|97.67|97.56|97.86|97.72|97.83|
> |Intra-Table Trends|95.38|94.99|95.58|95.34|95.42|
> |1-Hop|95.17|95.35|95.57|95.62|95.63|
> |**Runtime (min)**|**6**|**11**|**22**|**51**|**96**|
>
> ---
> ## [W3, Q3] Handling Hard Constraints
> In our framework, the database constraints are enforced through our graph-based representation of the RDB. Specifically, each row of every table is represented as a unique node in the graph, with its type defined by the table. The identity of each node in the graph can be used as its primary key in the database, **ensuring the uniqueness of primary keys**. On the other hand, foreign keys are represented as edges: each foreign key in the RDB corresponds to a directed edge from the child row's node to the parent row's node. Since edges are defined between nodes of different types (m-partite graph), **this ensures that foreign keys in one table correctly map to existing primary keys in another**.
> Our diffusion model is conditioned on the graph structure and only denoises the features; therefore, all these constraints are maintained during the diffusion process. Finally, given a generated graph, we get the RDB representation as follows (see Appendix D.1 for more details):
> 1. For each node type $i$, we assign a unique primary key $p_v \in \\{1,\dots,n_i \\}$ to each node $v$ of type $i$, where $n_i$ is the total number of nodes of type $i$. This ensures that nodes of the same type (i.e., from the same table) will have **unique primary keys**.
> 2. For each directed edge $(v_{child}, v_{parent})$ in the graph, we add $v_{parent}$'s primary key to the set of $v_{child}$'s foreign keys. This ensures that **foreign keys correctly map to existing primary keys**.
> 3. For each node type $i$, we construct the table $R^{(i)}$ as a collection of rows corresponding to all nodes of type $i$ in the graph, containing the primary keys, foreign keys, and the generated attributes.
>
> ---
> We hope our response can address the reviewer's concerns, and we are happy to answer any further questions during the discussion period.

---

### Official Review · Reviewer_s1Pb · 2025-07-02

**Clarity:** 2
**Significance:** 2
**Originality:** 2
**Rating:** 4
**Confidence:** 3

**Summary:**

This paper proposes Graph-Conditional Relational Diffusion Model (GRDM), a novel method that jointly modeling all tables in a relational database (RDB) without imposing any predefined order by adopting a natural graph representation of RDBs. GRDM leverages a graph neural network to simultaneously denoises row attributes and capture complex inter-table dependencies. Experimental results on 4 real-world graph datasets show some improvement against selected baseline methods (diffusion based), although for several cases the method underperforms SingleTable baseline.

**Questions:**

- In the ablation study, the performance gain from K=2 (2-hop) versus K=1 is very small. Can the authors explain why we need a multi-hop GNN in the final version? In other words, can we simply rely on diffusion to propagate information on the graph?
- Are there ways to verify the claims of preserving the privacy of the training database?

**Ethical Concerns:**

["NO or VERY MINOR ethics concerns only"]

**Final Justification:**

While I still find the evaluation on privacy as a weakness, I decided to increase my score to 4, given the limited time left and the author's argument that the generation task itself is challenging enough to outweigh privacy concerns.

**Limitations:**

yes

**Quality:**

3

**Strengths And Weaknesses:**

**Strength**

- The paper is moderately clear.
- The experimental results show improvement on the majority of the tasks (especially inter-table metrics)
- The proposed method is novel, where good modifications to a standard diffusion model for tabular data has been made, allowing it to generalize to multi-table settings.

**Weakness**

- While the paper claims data privacy as the major motivation for relational database generation, there is no analysis on how well the proposed method could preserve the privacy of the training dataset, making the effect of the proposed method questionable. This is especially concerning as the diffusion model may have simply memorized the training database, making the results concerning.
- Following the first point, the overall setting assumes that a different diffusion model is trained for different database, preventing a trained model to make meaningful generalizations. Moreover, this setting reinforces the concern where the model may have simply memorized the training database. There is no metric to ensure that memorization is not happening. As a thought experiment, if a model takes in a database, add some minor noises to the values, the model may achieve very high evaluation scores under the current setting as well.
- The method underperforms on intra-table metrics for 80% of the time compared to SingleTable baseline, it seems to sacrifice single table performance in some degree.
- The exact definition and implementations of the MP-GNN is not clear.

---

> ### Author Rebuttal · Authors · 2025-07-30
>
> We thank the reviewer for their valuable feedback. In the following, we address the reviewer's concerns in detail. We will include all the new experiments and discussions in the camera-ready version.
>
> ---
> ## [W1, W2, Q2] Privacy preservation and generalization
> In order to evaluate the privacy preservation of our model and to test if any memorization is happening, we follow prior work [1,2] and compute the distance to closest record (DCR) [3] between the generated and the training database. Specifically, for each synthetic sample, we get the minimum L1 distance to the real records. We report the mean DCR, the average of these distances over all generated samples. Since different tables can have different feature scales and ranges, we also report the DCR values from the holdout set to the training set, which serve as "ground-truth generalizations".
>
> We use 4 tables from different databases that contain sensitive information (but have been anonymized for these public benchmarks), such as census data containing household and individual information, sensitive financial information from a bank, and order information from an e-commerce platform. The results, provided in the table below, show that **our model yields higher DCR values than the baseline (i.e., is more private) across all tables** and it **even yields higher DCR values than the holdout set on 3 out of 4 tables**, while being close on the 4th one. These results suggest that **our model is able to generate new records from the underlying distribution and is not simply memorizing the training database**. Note that DCR values need to be considered along with fidelity metrics (see Table 1 in the paper), since, e.g., random noise can yield high DCR values.
>
> **Remark on W2:** While we use different diffusion models for different databases, each database contains multiple tables and potentially millions of rows, which are the central entity modeled by the same diffusion model. Note the contrast to single-table settings, where a different model is used for each table. While these single-table models showcased good levels of privacy protection and generalization [1,3], it is even more challenging for our model to memorize the training database because it models multiple tables at once.
>
> **Privacy metrics achieved by our model compared to the best baseline, and to the holdout set on 4 tables (the higher, the more private):**
> ||California - Household|California - Individual|Berka - Transaction|Instacart 05 - Order|
> |-|-|-|-|-|
> |Mean DCR - Holdout set|0.094|0.206|0.012|**0.012**|
> |Mean DCR - ClavaDDPM|0.104|0.177|0.019|0.007|
> |Mean DCR - GRDM (Ours)|**0.119**|**0.259**|**0.028**|0.009|
>
> ---
> ## [W3] Comparison to SingleTable baseline
> We designed our experiments such that all models share the same hyperparameters for the common components, including model architecture and size, diffusion steps, etc. In particular, the SingleTable baseline models tables separately and needs only to capture their individual distributions, while our model additionally needs to capture the correlations between different tables, given a similar parameter budget (the only difference is the GNN component, which constitutes approximately 5% of the total number of parameters).
>
> To better compare these two models, we provide the average single-table and multi-table metrics, respectively, in a new table below. This comparison shows that **the relative drop in single-table metrics is negligible compared to the relative gain in multi-table metrics achieved by our model**. Also, when averaged across all the databases, our model outperforms SingleTable on both single-table and multi-table metrics.
>
> In addition, we report the DCR values for the SingleTable baseline in the second table below (see response to [W1, W2, Q2] for the definition of this metric) and we can see that **the samples generated by SingleTable are consistently closer to the training data than our model, meaning that it generalizes less than our model**, which explains the higher single-table scores.
>
> **Comparison of average single-table and multi-table metrics between SingleTable and our model (the higher, the better):**
> ||SingleTable|GRDM (Ours)|Improvement(%)|
> |-|-|-|-|
> |**Berka**|
> |average single-table metrics|93.15|**97.56**|+4.73|
> |average multi-table metrics|78.47|**94.04**|+19.84|
> |**Instacart 05**|
> |average single-table metrics|94.5|**98.60**|+4.34|
> |average multi-table metrics|41.58|**94.10**|+126.31|
> |**RelBench-F1**|
> |average single-table metrics|95.87|**97.01**|+1.19|
> |average multi-table metrics|76.85|**95.28**|+23.98|
> |**MovieLens**|
> |average single-table metrics|**98.88**|97.23|-1.67|
> |average multi-table metrics|92.72|**95.60**|+3.11|
> |**CCS**|
> |average single-table metrics|**97.08**|95.32|-1.81|
> |average multi-table metrics|51.62|**85.54**|+65.71|
> |**California**|
> |average single-table metrics|**99.10**|98.58|-0.52|
> |average multi-table metrics|92.96|**97.68**|+5.08|
> |**Average across all databases**|
> |average single-table metrics|96.43|**97.38**|+0.99|
> |average multi-table metrics|72.37|**93.71**|+29.49|
>
> **Comparison of the privacy metrics between SingleTable and our model (the higher, the more private):**
> ||California - Household|California - Individual|Berka - Transaction|Instacart 05 - Order|
> |-|-|-|-|-|
> |Mean DCR - SingleTable|0.107|0.155|0.019|0.007|
> |Mean DCR - GRDM (Ours)|**0.119**|**0.259**|**0.028**|**0.009**|
>
> ---
> ## [W4] MP-GNN details
> In Appendix D.4, we provide a general formulation of heterogeneous MP-GNNs. In the following, we provide the exact equations of the heterogeneous GraphSAGE model, which we used in our experiments, by plugging in the concrete choices of the different model components (e.g. choice of the aggregation function). Our implementation is based on the official PyTorch Geometric (PyG) implementation of the heterogeneous GraphSAGE model.
>
> We describe one layer of this model, which maps node embeddings $\\{h_v^l \\} _{v \in \mathcal{V}}$ to $\\{h_v^{l+1} \\} _{v \in \mathcal{V}}$, where $l$ denotes the layer count. We describe the GNN update for a single node, and the same operations are applied to all nodes in parallel.
> 1. **Compute a message for each edge type**. To accommodate the heterogeneity of the graph, a distinct message is computed for each edge type. For each edge type $e$, $\mathcal{N}_ e (v)$ denotes the neighbors of $v$ that are connected via the edge type $e$, and $g_e$ is a neural network (typically an MLP) that is specific to this edge type. Then, the following message is computed for each edge type $e$: $$m_{v,e}^{l+1}=\sum_{w\in\mathcal{N}_e(v)}g_e(h_w^l).$$
> 2. **Combine the messages into a unified message**. Let $f_e$ be another MLP that processes messages of type $e$. Then, the combined message is computed as: $$m_v^{l+1}=\sum_ef_e(m_{v,e}^{l+1}).$$
> 3. **Update node embedding**. The unified message is used to update the node embedding. Again, to accommodate the heterogeneity of the graph, different MLPs are applied depending on the node's type, which we denote as $\phi(v)$. The updated node embedding is: $$h_v^{l+1}=f_{\phi(v)}(h_v^l,m_v^{l+1}).$$
>
> These computations are run for all nodes, defining a single layer. Multiple such layers can be stacked to map initial node embeddings $\\{h_v^0 \\} _{v \in \mathcal{V}}$ to $\\{h_v^L \\} _{v \in \mathcal{V}}$, where $L$ is the total number of layers, with different neural networks for each layer. For each node $v$, $h_v^L$ contains information about the node $v$ and its $L$-hop neighborhood and can be used for downstream processing, e.g., type-specific denoising.
>
> ---
> ## [Q1] K=2 vs K=1
> Indeed, the K=1 version achieves very strong performance and already outperforms all the baselines on the inter-table metrics, which confirms that we can safely rely on diffusion to propagate information on the graph (as long as we use K>0).
> The multi-hop version can be seen as a simple way to further improve the performance at the cost of some additional compute.
>
> ---
> ## References
> [1] Kotelnikov, Akim, et al. Tabddpm: Modelling tabular data with diffusion models.
>
> [2] Pang, Wei, et al. Clavaddpm: Multi-relational data synthesis with cluster-guided diffusion models.
>
> [3] Zhao, Zilong, et al. Ctab-gan: Effective table data synthesizing.
>
> ---
> We hope our response can address the reviewer's concerns, and we are happy to answer any further questions during the discussion period.

---

> > ### Comment · Reviewer_s1Pb · 2025-08-06
> >
> > I appreciate the author's response that addresses some of my concerns regarding performance and clarity. However, the concerns on privacy still hold, as the provided privacy evaluation seems rather simplistic as it only considers the L1 distance instead of protections under identifiability attacks. It is also focused on comparison with diffusion-based methods only, while there has been rich literature on synthetic tabular data with other generative models where comprehensive discussions on privacy have been done (see list of privacy metrics in https://github.com/vanderschaarlab/synthcity, and https://github.com/schneiderkamplab/syntheval).
> > - How does the work situate itself against other works in the tabular data synthesis tasks besides diffusion and autoregressive?
> > - Can privacy concerns be addressed with more principled approaches, e.g., with differential privacy guarantees?

---

> > > ### Author Response · Authors · 2025-08-07
> > > **Clarifications regarding privacy**
> > >
> > > We thank the reviewer for their response, and we are glad that our rebuttal addressed their concerns regarding performance and clarity. Regarding privacy, we would like to make the following clarifications:
> > >
> > > * First, we would like to clarify that privacy is not the main focus of this work. While we present privacy as one of the major motivations for building generative models for relational databases, our focus in this paper is on building generative models capable of capturing the complexity of real-world multi-table databases, which existing methods fail to achieve (see our main results in Table 1). We view this as a prerequisite for achieving the ultimate goal of private synthetic data, since generating private data that does not capture the complexity of the real data is trivial but useless (e.g., random noise). Extending our framework with formal privacy certificates such as differential privacy is a very interesting research direction, which we plan to tackle in future work. However, this direction is technically orthogonal to the contributions of this work.
> > >
> > > * The main contribution of this work is on the modeling side. Specifically, we introduce the first non-autoregressive generative model for relational databases and show that it significantly outperforms existing methods on capturing the correlations between different tables in a database, achieving up to a 5x improvement (see Table 1).
> > >
> > > * In our rebuttal, we provided some preliminary results on the privacy preservation of our model, which show that it makes meaningful generalizations beyond simply memorizing the training data. The L1/L2 distance between the synthetic and the real data is a widely used sanity check used by many recent works on relational and tabular data generation, see e.g. [1,2,3,4,5].
> > >
> > > * Note that the links shared by the reviewer focus on single-table generation, which is a much more mature but also simpler research area than multi-table relational data generation, where current models still struggle to even capture the complexity of the data (see first and second points). However, our proposed framework is general enough to be applied with different generative models backbones, including differentially private ones.
> > >
> > > We hope that these points help clarify the scope of our work. We are happy to answer any further questions. We sincerely hope that the reviewer will reconsider their evaluation of our work.
> > >
> > > ---
> > >
> > > ### References
> > > [1] Pang, Wei, et al. "Clavaddpm: Multi-relational data synthesis with cluster-guided diffusion models." Advances in Neural Information Processing Systems 37 (2024): 83521-83547.
> > >
> > > [2] Hudovernik, Valter, et al. "RelDiff: Relational Data Generative Modeling with Graph-Based Diffusion Models." arXiv preprint arXiv:2506.00710 (2025).
> > >
> > > [3] Hudovernik, Valter, Martin Jurkovič, and Erik Štrumbelj. "Benchmarking the fidelity and utility of synthetic relational data." arXiv preprint arXiv:2410.03411 (2024).
> > >
> > > [4] Kotelnikov, Akim, et al. "Tabddpm: Modelling tabular data with diffusion models." International conference on machine learning. PMLR, 2023.
> > >
> > > [5] Zhang, Hengrui, et al. "Mixed-type tabular data synthesis with score-based diffusion in latent space." arXiv preprint arXiv:2310.09656 (2023).

---

> > > > ### Comment · Reviewer_s1Pb · 2025-08-08
> > > >
> > > > Thanks for the clarification. While I still don't see why the more established privacy metrics for tabular data can't be used in this task, given that L1 distance is computable, I am open to revising my score given the limited time left and the author's argument that the generation task itself is challenging enough to outweigh privacy concerns.

---

> > > > > ### Author Response · Authors · 2025-08-09
> > > > > **Thank you**
> > > > >
> > > > > We sincerely thank the reviewer for taking our clarification into account and for their openness to revise their evaluation of our work.

---

### Official Review · Reviewer_gCht · 2025-07-07

**Clarity:** 3
**Significance:** 3
**Originality:** 2
**Rating:** 4
**Confidence:** 4

**Summary:**

This paper proposes GRDM, a novel non-autoregressive model for synthetic relational database generation. GRDM represents databases as graphs and uses diffusion models to jointly generate all attributes. The method captures complex inter-table dependencies without requiring table ordering. Experiments show strong results across multiple real-world datasets.

**Questions:**

How would GRDM perform if some tables were missing or incomplete at test time?

Can the graph generation be made differentiable to allow end-to-end training?

Does the performance drop when increasing database size?

**Ethical Concerns:**

["NO or VERY MINOR ethics concerns only"]

**Final Justification:**

Appreciate authors' response and have raised my score.
Thanks again for the hard work and well executed response.

**Limitations:**

yes

**Paper Formatting Concerns:**

/

**Quality:**

3

**Strengths And Weaknesses:**

Strengths
-- Nice work. It avoids autoregressive assumptions, allowing parallel generation and better scalability.
-- Captures long-range dependencies via graph-based diffusion.

Weaknesses:
-- Graph generation step is quite simple and may limit structure realism. Consider more advanced generative graph models to improve relational structure fidelity.
-- Categorical data is encoded as continuous using label encoding, which may harm semantics. Future work could explore discrete diffusion or hybrid encodings to better handle categorical variables.
-- Unclear how well the model handles relational schema changes or dynamic schemas. Adding a discussion or experiments on generalization to modified schemas would improve the work.
-- Missing related works: Check the papers by V. Hudovernik. Especially his RelDiff and benchmark against it

---

> ### Author Rebuttal · Authors · 2025-07-30
>
> We thank the reviewer for their valuable feedback. In the following, we address the reviewer's concerns in detail. We will include all the new experiments and discussions in the camera-ready version.
>
> ---
> ## [W1] Evaluation of the graph generation step
>
> To better understand the modeling capacity of our graph generation algorithm, we perform a new ablation by replacing the sampled graph with the ground-truth graph and using the same diffusion model used in the paper to generate the attributes. The ground-truth graph achieves a perfect relational structure fidelity (cf. Cardinality metric) and provides an upper-bound on the performance achievable by graph generative models.
> The results in the following table show that our model achieves **very similar performance to the ground-truth graph setting**, showing that **our graph generation procedure can effectively capture the relational structure of the RDBs**.
> We also note that existing deep generative models for graphs are limited to graphs with a few thousand nodes, and scaling them to relational graphs with millions of nodes is a very interesting research direction.
>
> ||Sampled Graph (Ours)|Ground-truth Graph
> |-|-|-|
> |**Berka**
> |Cardinality|99.70 ± 0.07|100.0 ± 0.00
> |Column Shapes|96.90 ± 0.07|97.00 ± 0.14
> |Intra-Table Trends|98.21 ± 0.05|98.24 ± 0.06
> |1-Hop|92.94 ± 0.06|93.16 ± 0.02
> |2-Hop|96.04 ± 0.20|96.67 ± 0.14
> |3-Hop|93.13 ± 0.47|94.47 ± 0.94
> |**Movie Lens**
> |Cardinality|98.80 ± 0.36|100.0 ± 0.00
> |Column Shapes|98.22 ± 0.05|98.25 ± 0.10
> |Intra-Table Trends|96.24 ± 0.24|96.31 ± 0.15
> |1-Hop|95.60 ± 0.17|94.19 ± 2.12
> |**California**
> |Cardinality|99.96 ± 0.01|100.0 ± 0.00
> |Column Shapes|99.15 ± 0.02|99.15 ± 0.01
> |Intra-Table Trends|98.00 ± 0.01 |98.02 ± 0.02
> |1-Hop|97.68 ± 0.01 |97.68 ± 0.02
>
> ---
> ## [W2] Discrete diffusion for categorical variables
> In the early stages of the project, we experimented with multinomial diffusion [1] for categorical variables, which is based on uniform transitions between discrete states. However, we found it to **perform significantly worse than unified Gaussian diffusion**, as can be seen from the table below. This is likely because discrete diffusion models like uniform and masked diffusion completely **ignore the relationships between the different states** that can be important in, e.g., ordinal data, which is abundant in real-world RDBs (counts, discrete timesteps, ratings, etc.). On the other hand, Gaussian diffusion leverages these relationships by transitioning between more similar states with higher probability.
> We also compare to concurrent work, RelDiff [2], which uses masked diffusion, and show that our model outperforms it on most datasets (see response to W4).
>
> ||Multinomial + Gaussian diffusion|Unified Gaussian diffusion
> |-|-|-|
> |**California**
> |Column Shapes|83.27 ± 0.07|**99.51 ± 0.04**
> |Intra-Table Trends|79.27 ± 0.08|**98.69 ± 0.08**
> |**Berka**
> |Column Shapes|76.41 ± 2.21|**94.58 ± 0.01**
> |Intra-Table Trends|72.80 ± 1.23|**91.72 ± 0.23**
>
> ---
> ## [W3] Generalization to modified schemas
> While we are not aware of any other model that can handle schema changes, we discuss how our model can be adapted to handle different schemas:
> 1. First, we need a collection of databases with different schemas to enable model generalization. We could use different databases and/or create different schemas of the same database using schema change operations.
> 2. To enable training on different schemas, the generation process needs to be conditioned on the schema. For example, we could model the schema as a graph (nodes=tables, edges=table-level connections) and use an encoding of this graph as input to the generation process.
> 3. Schema changes typically involve data type modifications, so we need a unified representation for different data types. We could use tabular foundation models pre-trained on large amounts of data to map different data types onto a unified latent space that can be used for generation.
> 4. With these changes, we can apply our framework to jointly model the rows of any database in a unified space conditioned on its schema. After training on a variety of schemas, the model should be able to generalize to new schemas.
>
> Due to the limited time during the rebuttal, we leave this exciting direction for future research.
>
> ---
> ## [W4] Comparison to RelDiff and RGCLD
> Unfortunately, RelDiff [2] first appeared online 2 weeks after the NeurIPS full paper submission deadline, and at the time of writing, their code is still not available. Nonetheless, we provide a comparison to RelDiff on the datasets and metrics that are common to our evaluation setup. We also include RGCLD [3], which generates tables autoregressively. The results show that **GRDM significantly outperforms both models on the vast majority of the metrics**.
>
> ||RGCLD|RelDiff|GRDM (Ours)
> |-|-|-|-|
> |**Berka**
> |Cardinality|**100.0**|**100.0**|99.70 ± 0.07
> |Column Shapes|79 ± 0.04|**97.72 ± 0.03**|96.90 ± 0.07
> |Intra-Table Trends|74 ± 0.03|**98.81 ± 0.02**|98.21 ± 0.05
> |1-Hop|81 ± 0.03|**96.88 ± 0.06**|92.94 ± 0.06
> |2-Hop|74 ± 0.03|95.79 ± 0.02|**96.04 ± 0.20**
> |3-Hop|65 ± 0.09|90.19 ± 0.22|**93.13 ± 0.47**
> |**Instacart 05**
> |Cardinality|N.A.|**100.0**|99.96 ± 0.01
> |Column Shapes|N.A.|96.85 ± 0.85|**98.87 ± 0.06**
> |Intra-Table Trends|N.A.|95.71 ± 0.42|**98.33 ± 0.04**
> |1-Hop|N.A.|85.83 ± 1.20|**92.03 ± 0.30**
> |2-Hop|N.A.|70.74 ± 0.14|**96.17 ± 0.15**
> |**RelBench-F1**
> |Cardinality|**100.0**|**100.0**|98.26 ± 0.12
> |Column Shapes|91 ± 0.03|94.89 ± 0.08|**97.28 ± 0.29**
> |Intra-Table Trends|92 ± 0.01|95.10 ± 0.12|**96.74 ± 0.36**
> |1-Hop|88 ± 0.03|93.46 ± 0.10|**93.74 ± 0.50**
> |2-Hop|88 ± 0.03|95.91 ± 0.03|**96.81 ± 0.28**
> |**MovieLens**
> |Cardinality|**100.0**|**100.0**|98.80 ± 0.36
> |Column Shapes|94 ± 0.05|96.91 ± 0.49|**98.22 ± 0.05**
> |Intra-Table Trends|94 ± 0.04|93.88 ± 0.87|**96.24 ± 0.24**
> |1-Hop|86 ± 0.09|94.84 ± 0.35|**95.60 ± 0.17**
>
> ---
> ## [Q1] Missing and incomplete tables at test time
> To assess the performance of GRDM in the case of missing or incomplete tables at test time, we design a new experiment on the California database.
> First, we split the database into a train and a holdout set based on the parent table, and use the train set to train the diffusion model in the same way as presented in the paper. Next, we perform experiments on the holdout set by conditioning the generation on some sections and using the same pre-trained model to inpaint the missing sections without any modification [4]. We report the metrics comparing the generated and the holdout set. We use 3 different strategies to select the missing parts of the database:
>
> 1. Masking entire tables.
> 2. Masking entire rows from all tables with different rates.
> 3. Masking single cells (individual attributes of rows) from all tables with different rates.
>
> The results show that **GRDM consistently maintains good performance across the different conditional generation settings**.
>
> ||Unconditional|Parent table is missing|Child table is missing|50% of rows are missing|75% of rows are missing|50% of cells are missing|75% of cells are missing
> |-|-|-|-|-|-|-|-|
> |Cardinality|99.78|100.0|100.0|100.0|100.0|100.0|100.0
> |Column Shapes|99.03|99.26|99.37|99.22|99.05|98.70|98.10
> |Intra-Table Trends|97.75|98.65|98.56|98.36|97.91|96.34|94.44
> |1-Hop|97.34|96.48|97.16|97.66|97.17|97.20|95.96
>
> ---
> ## [Q2] Differentiable graph generation
> Yes, the graph generation can be made differentiable. For instance, we can use another diffusion model to generate the graph structure. However, under our proposed factorization $p(\mathcal{G})=p(\mathcal{V},\mathcal{E})p(\mathcal{X}|\mathcal{V},\mathcal{E})$, typically the graph generative model and the conditional feature generative model are trained separately, using teacher forcing for the conditional model, similarly to how autoregressive models are trained in practice.
> To allow end-to-end training, we would need to directly model the joint $p(\mathcal{G})=p(\mathcal{V},\mathcal{E},\mathcal{X})$, which can pose computational challenges when scaling to very large graphs (see section 3.2 for a detailed discussion).
>
> ---
> ## [Q3] Performance when increasing database size
> Our graph generation algorithm allows us to generate graphs of arbitrary sizes. We leverage this property to test the performance of GRDM when increasing the database size. We use different multipliers of the original database size and generate databases with up to more than **20 million rows** (10x multiplier) using the same diffusion model. The results, shown below, highlight that **the performance does not drop when increasing the database size**. This can be explained by the use of our GNN in the denoising model, which operates on local subgraphs that are roughly invariant to the overall size of the RDB. Note that all RDBs were generated within less than 2 hours.
>
> |Size multiplier (number of rows)|0.5x (1.037M)|1x (2.076M)|2x (4.154M)|5x (10.381M)|10x (20.755M)
> |-|-|-|-|-|-|
> |**California**
> |Cardinality|99.89|99.96|99.96|99.99|99.97
> |Column Shapes|97.67|97.67|97.69|97.68|97.67
> |Intra-Table Trends|95.12|95.12|95.15|95.14|95.14
> |1-Hop|94.67|94.64|94.64|94.63|94.64
>
> |Size multiplier (number of rows)|0.5x (0.588M)|1x (1.221M)|2x (2.501M)|5x (6.270M)|10x (12.414M)
> |-|-|-|-|-|-|
> |**Movie Lens**
> |Cardinality|98.69|98.87|99.10|99.63|99.65
> |Column Shapes|97.67|97.56|97.86|97.72|97.83
> |Intra-Table Trends|95.38|94.99|95.58|95.34|95.42
> |1-Hop|95.17|95.35|95.57|95.62|95.63
>
> ---
> ## References
> [1] Kotelnikov, Akim, et al. Tabddpm: Modelling tabular data with diffusion models.
>
> [2] Hudovernik, Valter, et al. RelDiff: Relational Data Generative Modeling with Graph-Based Diffusion Models.
>
> [3] Hudovernik, Valter. Relational data generation with graph neural networks and latent diffusion models.
>
> [4] Lugmayr, Andreas, et al. Repaint: Inpainting using denoising diffusion probabilistic models.
>
> ---
> We hope our response can address the reviewer's concerns, and we are happy to answer any further questions during the discussion period.

---

### Note · Authors · 2025-08-16

We thank the reviewers again for their valuable feedback, which helped us improve the paper. We proposed GRDM, a diffusion model for relational database (RDB) generation that leverages a graph representation of RDBs and jointly generates their attributes, avoiding the limitations of autoregressive models. The reviewers appreciated the novelty and soundness of our idea and the clarity of the paper. They were generally positive about the experimental setup, which highlights the strong performance of our model in capturing long-range dependencies in RDBs.

The reviewers also raised valid concerns about some aspects of our work. We leveraged their feedback to clarify more details, conduct new experiments, and discuss possible extensions. In the following, we recap their main concerns and summarize our responses:
* Privacy preservation: We conducted a new privacy analysis of our model, showing that it does not memorize the training data and is able to generate new records from the underlying distribution. The reviewer (s1Pb) appreciated our clarifications and said they were open to revising their evaluation of our work.
* Graph generation step: We conducted a new ablation by replacing the generated graph with the ground-truth graph and found similar results in both cases. We also reported new higher-order structural metrics. Both experiments showed that our algorithm is able to effectively capture the complex structural properties of real-world RDBs.
* Scalability and generalization: To test the scalability of our model, we generated synthetic RDBs with up to 10x the size of the original RDB and found that the model maintained very good performance across different scales. We also designed a new experiment to test our model in the task of missing value imputation. Furthermore, we discussed how our model can be modified to generate RDBs with different schemas.
* Discrete diffusion and additional baselines: To justify our use of Gaussian diffusion for categorical data, we provided a new comparison between unified Gaussian diffusion and hybrid diffusion, and found that the former performs significantly better. We also compared our model to additional baselines and found our model to perform significantly better overall.

Based on the feedback we received from some reviewers, we believe that we have addressed most concerns. We are fully committed to including the new experiments, clarifications, and discussions in the final manuscript, in case of acceptance.

---

### Decision · Program_Chairs · 2025-09-17

**Decision:**

Accept (poster)

**Comment:**

This paper studies the limitations of prior RDB generative models—focusing on single tables or using autoregressive sequential generation. The authors propose GRDM, a graph-conditional relational diffusion model that jointly models all RDB tables via graph neural networks to denoise rows and capture inter-table dependencies. The authors and reviewers had thorough rebuttal discussions, and finally, the reviewers converged to consistently accept this paper. After reading the paper & discussions, I think this paper is acceptable for publication in NeurIPS this year.